# The genomic footprint of whaling and isolation in fin whale populations

Sergio F. Nigenda-Morales [1,12,15] ✉, Meixi Lin [2,13,15] ✉,
Paulina G. Nuñez-Valencia [1,3], Christopher C. Kyriazis[2],
Annabel C. Beichman [4], Jacqueline A. Robinson [5], Aaron P. Ragsdale[1,14],
Jorge Urbán R. [6], Frederick I. Archer [7], Lorena Viloria-Gómora[6],
María José Pérez-Álvarez [8,9], Elie Poulin [9], Kirk E. Lohmueller[2,10,11] ✉,
Andrés Moreno-Estrada [1] ✉ & Robert K. Wayne[2,16]

Twentieth century industrial whaling pushed several species to the brink of extinction, with fin whales being the most impacted. However, a small, resident population in the Gulf of California was not targeted by whaling. Here, we analyzed 50 whole-genomes from the Eastern North Pacific (ENP) and Gulf of California (GOC) fin whale populations to investigate their demographic history and the genomic effects of natural and human-induced bottlenecks. We show that the two populations diverged ~16,000 years ago, after which the ENP population expanded and then suffered a 99% reduction in effective size during the whaling period. In contrast, the GOC population remained small and isolated, receiving less than one migrant per generation. However, this low level of migration has been crucial for maintaining its viability. Our study exposes the severity of whaling, emphasizes the importance of migration, and demonstrates the use of genome-based analyses and simulations to inform conservation strategies.

Due to increasing recent human impacts, many vertebrate species have experienced drastic population declines and now persist as small and fragmented populations[1–3]. Small populations are at higher risk of population declines due to stochastic environmental and genetic factors[4–6]. Both anthropogenic and naturally occurring population declines reduce genetic diversity, and increase inbreeding and genetic load due to the stronger action of genetic drift which diminish the long-term survival and adaptive potential of populations[7,8]. However, the impact of these processes depends on the often unknown population-specific demographic histories and life history traits. For example, gene flow as low as one effective migrant per generation may counteract genetic drift and reduce the frequency of deleterious

[1]Advanced Genomics Unit, National Laboratory of Genomics for Biodiversity (Langebio), Center for Research and Advanced Studies (Cinvestav), Irapuato, Guanajuato 36824, Mexico. [2]Department of Ecology and Evolutionary Biology, University of California, Los Angeles, Los Angeles, CA 90095, USA. [3]Centro de Ciencias Genómicas, Universidad Nacional Autónoma de México (UNAM), Cuernavaca, Morelos, México. [4]Department of Genome Sciences, University of Washington, Seattle, WA 98195, USA. [5]Institute for Human Genetics, University of California, San Francisco (UCSF), San Francisco, CA 94143, USA. [6]Departamento de Ciencias Marinas y Costeras, Universidad Autónoma de Baja California Sur (UABCS), La Paz, Baja California Sur, Mexico. [7]Marine Mammal and Turtle Division, Southwest Fisheries Science Center, La Jolla, CA 92037, USA. [8]Escuela de Medicina Veterinaria, Facultad de Medicina y Ciencias de la Salud, Universidad Mayor, Santiago, Chile. [9]Millennium Institute Biodiversity of Antarctic and Subantarctic Ecosystems (BASE), Universidad de Chile, Santiago, Chile. [10]Interdepartmental Program in Bioinformatics, University of California, Los Angeles, CA 90095, USA. [11]Department of Human Genetics, David Geffen School of Medicine, University of California, Los Angeles, CA, USA. [12]Present address: Department of Biological Sciences, California State University San Marcos, San Marcos, CA 92096, USA. [13]Present address: Department of Plant Biology, Carnegie Institution for Science, Stanford, CA 94305, USA. [14]Present address: Department of Integrative Biology, University of Wisconsin, Madison, WI 53706, USA. [15]These authors contributed equally: Sergio F. Nigenda-Morales, Meixi Lin. [16]Deceased: Robert K. Wayne. ✉e-mail: snigenda@csusm.edu; meixilin@ucla.edu; klohmueller@g.ucla.edu; andres.moreno@cinvestav.mx

variation[9–11], but might also reduce metapopulation genetic variation[12], or introduce strongly deleterious alleles[13]. Therefore, uncovering population history and determining how detrimental genetic patterns arise in declining populations are challenging questions, but the answers are critical to developing effective conservation strategies[14].

Industrial whaling during the 20th century is arguably one of the most disruptive ecological events caused by humans[15], which decimated all great whale species and drove many of them to the brink of extinction[16,17]. Estimating the decline of whale populations is crucial to evaluate the full impact of whaling and designing appropriate recovery policies, not only on whale abundance but on entire ecosystems[15,17,18]. However, quantifying the magnitude of known recent population declines in endangered vertebrate species from contemporary samples has proven difficult because the estimates based on genetic diversity capture long-term effective sizes rather than recent demographic events[19,20]. Additionally, the long life span and generation time of whales complicate the inference of recent population size changes[21] because less generation turnover occurs in a given amount of time. Given these challenges, previous genetic studies using contemporary samples have only indirectly inferred the impact of whaling by determining that historical abundance estimates obtained from whaling records and ecological studies are orders of magnitude lower than those based on the diversity of a few mitochondrial or nuclear markers[17–24], suggesting a slower recovery of whale populations after the end of whaling. Therefore, to overcome these challenges, we used high-coverage whole-genome sequence data and model-driven approaches to provide more power and resolution to directly detect recent demographic changes[19,25], such as whaling.

The fin whale (*Balaenoptera physalus*) is the second-largest whale and the one most impacted by industrial whaling worldwide. In the North Pacific alone, more than 75,500 fin whales were harvested[26]. However, fin whales in the Gulf of California, Mexico, belong to a resident population that was not targeted by whalers[27,28]. Nevertheless, this population has been small with limited gene flow from and to the Pacific for thousands of years[28–31]. In contrast, the Eastern North Pacific population was large, interconnected, and overexploited[27], although the population along the U.S. west coast has shown evidence of growth at 3% per year since the 1990's[32].

Here, we provide direct genome-wide demographic reconstructions of whaling in a previously large population, in comparison to a never-whaled but small and isolated population. We analyze and model the whole-genome diversity of fin whale populations with contrasting demographic histories to identify the genetic and evolutionary impacts of population reductions in large, long-lived marine mammals. Understanding the complex interaction between demographic and evolutionary factors shaping the genetic diversity in whale populations is key to improving their conservation, especially given current and future whaling threats and the challenges of climate change and human inputs to marine ecosystems[17]. Evaluating the genomic consequences of contrasting population reductions in fin whale populations make our results relevant for the conservation of populations in other threatened or endangered species.

## Results

### Sampling, population structure, and differentiation

To assess the genome-wide impact of human-induced and natural bottlenecks on fin whale populations, we generated high coverage (average 27×) whole-genome resequencing data from 50 samples of free-ranging individuals collected between 1995 and 2017 (Fig. 1A; Table S1). Thirty individuals are from regions that survived intensive whaling pressure in the Eastern North Pacific (ENP), along the coasts of California (CA; $N = 9$), Oregon (OR; $N = 4$), Washington (WA; $N = 2$), British Columbia (BC; $N = 3$) and Alaska (AK; $N = 12$). Additionally, we included 20 individuals from a naturally small population in the Gulf of California, Mexico (GOC), that has maintained a low population size

between 300 and 600 individuals for thousands of years and avoided the impacts of whaling[27,30,31].

The sequences were aligned, genotyped, annotated and filtered using the minke whale genome as a reference (BalAcu1.0). We also genotyped a subset of ten individuals using a recently available fin whale genome assembly (GCA_023338255.1). We observed only a 1.5% overestimation of diversity when using the minke whale genome as reference, which could be due to a less accurate mapping (See Supplemental Discussion). Also, both reference genomes provide similar genotyping statistics and genomic diversity results (Table S2; Fig. S1; Supplemental Methods and Results), suggesting that using the minke whale genome as a reference does not introduce significant biases in our analyses (see discussion and significance tests in Supplemental Results and Discussion). Principal component analysis (PCA) separated the ENP and GOC individuals on PC1 with tight clustering of the GOC samples (Fig. 1B). A wider dispersion pattern is observed for the ENP samples, with the Alaska samples remaining relatively clustered, suggesting some degree of differentiation of this northern population from those to the south (Fig. S2). Admixture analysis of all the samples supports a $K = 2$ partition of ENP and GOC samples (Figs. 1C, S3). We identified one ~50% admixed individual from each population (ENPCA09 and GOC010) and a small admixture fraction from GOC in the ENP population (Fig. 1B, C). Additional admixture analysis of only ENP samples supports a $K = 1$ partition of this population (Fig. S4). $F_{ST}$ values are higher between the GOC and ENP ($F_{ST} = 0.073$, $p = 0.001$) than between all locations within the ENP ($F_{ST} = 0$–$0.008$; Table S3). Assuming the highest $F_{ST}$ of 0.008 observed within ENP, this substructure would at most inflate effective population size ($N_e$) estimates by 0.8%[33]. Also, a phylogenetic analysis separated both populations into different groups, with the nodes within the ENP group showing no bootstrap support. The two admixed individuals clustered with ENP but showed early divergence (Fig. S5), suggesting their greater genetic differentiation. These results indicate there are two main populations in our sample, one off the Pacific coast and the other in the Gulf of California, consistent with previous microsatellite and mitochondrial data[30,31]. In addition, our findings confirm the strong isolation of the geographically distinct Gulf population[30,34], whereas weak population substructure was observed in the eastern North Pacific.

### Genome-wide patterns of variation and runs of homozygosity

We explored the genome-wide diversity patterns of fin whale populations by calculating average genome-wide heterozygosity and per-site heterozygosity in non-overlapping 1-Mb windows. In GOC individuals we found patterns of reduced variation, with an average 1.13 heterozygotes per kb (het/kb) and an increased proportion of genomic regions with low heterozygosity (46% of windows contain <1 het/kb). In contrast, the ENP population had much higher diversity (1.76 het/kb; two-tailed Mann–Whitney U [MWU] test $p = 1.15\text{E-}10$; Fig. 2A) and few regions of low heterozygosity (12% of windows with <1 het/kb; Figs. 2B, S6, S7). These genome-wide results imply contrasting demographic histories of long-term small and large population size in the Gulf and North Pacific, respectively[30]. Compared with other cetaceans that experienced different levels of population contractions, such as the diminutive vaquita porpoise (*Phocoena sinus*) in the Gulf of California[35,36] (0.1 het/kb), abundant minke whale[37] (0.6 het/kb) and endangered blue whale[38] (2.1 het/kb), the GOC fin whales have maintained moderate genome-wide patterns of variation (Fig. 2A), suggesting that evolutionary mechanisms such as migration have maintained genetic diversity. However, the GOC population has an enriched number of 1-Mb windows with null or very low heterozygosity (0–0.1 het/kb) compared with more endangered mysticete species such as the North Atlantic right whale and blue whale (Fig. S8), indicating that populations of these endangered species were historically larger than the Gulf of California fin whale population and imply a

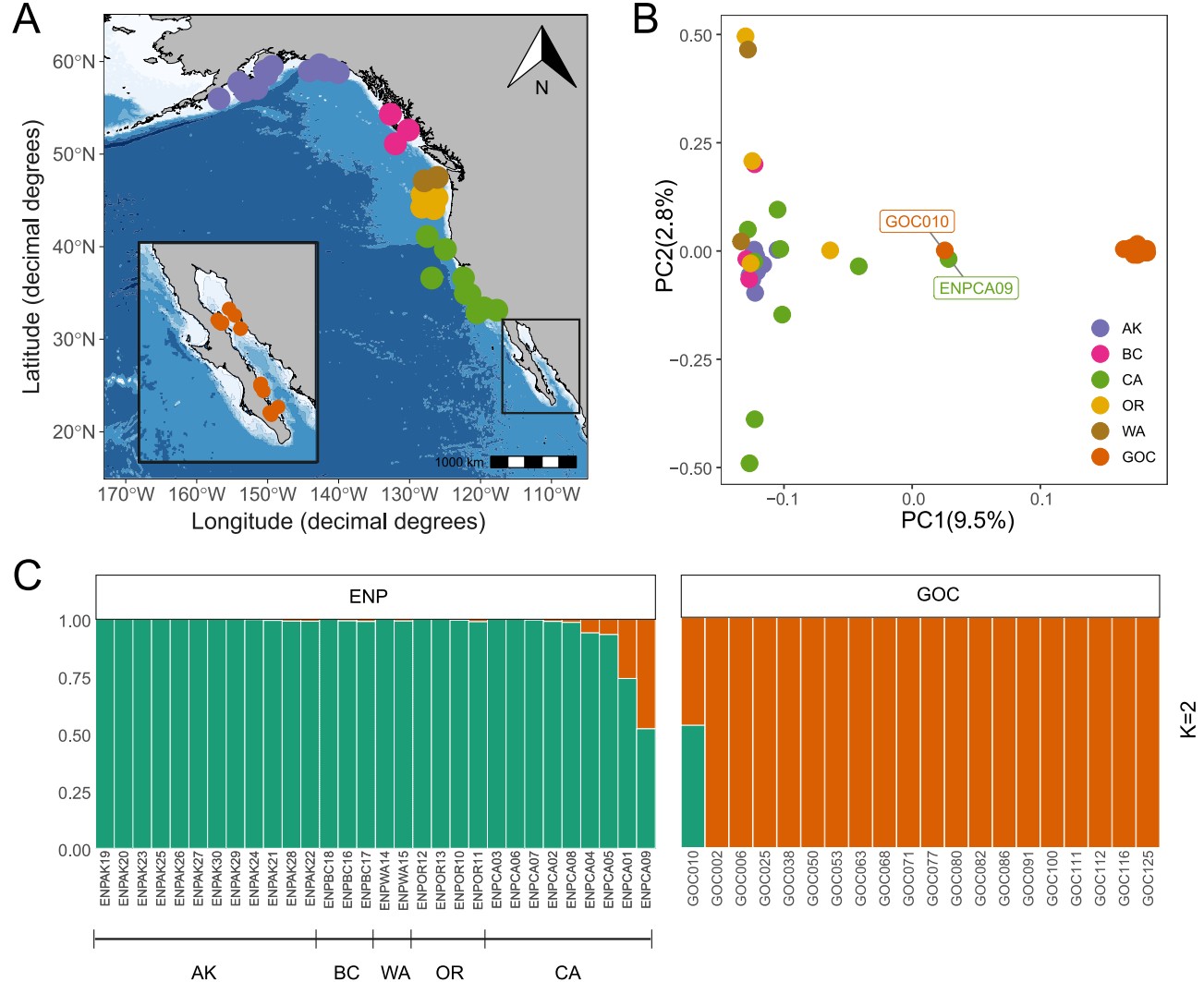

**Fig. 1 | Population structure and sample origins for the fin whale genomes obtained in this study. A** Thirty skin samples were collected along Eastern North Pacific (ENP) locations near Alaska (AK), British Columbia (BC), Washington (WA), Oregon (OR), and California (CA) from 1995 to 2017. Twenty samples were collected in seven sites within the Gulf of California (GOC) from Bahía de La Paz and Los Frailes in the southern Gulf to Bahía de los Ángeles, Puerto Refugio, and Bahía Kino around the Midriff islands (Table S1). **B** PCA for 50 samples are colored by their location origin. The admixed individuals are labeled. **C** Admixture analyses supported two ancestral populations (K = 2). The map in **A** was generated with the R package ggOceanMaps[112] which uses publicly available bathymetry data from the ETOPO1 1-arc minute global relief data set distributed by the National Center for Environmental Information[113] (https://www.ncei.noaa.gov/products/etopo-global-relief-model). Source data are provided as a Source Data file.

reassessment towards a more threatened status of the GOC population may be needed.

To characterize the history of inbreeding events, we identified runs of homozygosity (ROH), which are genomic stretches within an individual that are assumed to be identical by descent, using two model-based methods[39,40] (Fig. S9). Long ROH (≥5 Mb) typically result from recent close inbreeding whereas shorter ROH indicate either older inbreeding or older reductions in population size[41]. Overall, GOC individuals contained considerably more ROH segments than ENP individuals (two-tailed MWU test $p = 9.42E{-}08$), but most of the ROH were of short (0.1–1 Mb) or intermediate (1–5 Mb) length (Fig. 2A). Long ROH were present in all GOC individuals, except the admixed sample GOC010, and only in three ENP individuals. Nevertheless, they comprise a small fraction of total ROH length in both populations ($F_{ROH \geq 5M} = 0.4{-}3.1\%$; Table S4). To further explore the timing of inbreeding, we estimated the average time at which two homologous haplotypes could coalesce within our ROH categories for each population, assuming a recombination rate of 1 cM/Mb[42]. For short ROH, haplotypes coalesced on average approximately 145 and 250

generations ago in GOC and ENP, respectively, whereas for intermediate ROH the average haplotype coalescent time was 28 and 30 generations ago. These findings suggest a lack of recent inbreeding in both populations (Figs. 2A, S10). However, the higher number and longer ROH observed in the GOC fin whales (Figs. 2A, S9, S10), together with the high proportion of their genome contained in ROH larger than 1 Mb ($F_{ROH \geq 1M(GOC)} = 17.5{-}23.4\%$; Table S4), indicate that genomic segments in this population share a more recent common ancestor than they do in the Pacific population. Finally, we determined the relatedness between individuals in both populations and found significantly higher average kinship coefficient among GOC individuals (0.054) than in the ENP population (0.0032; two-tailed MWU test $p < 2.2E{-}16$), indicating greater identity-by-descent in the GOC, which further demonstrate higher inbreeding levels in this population (Fig. S11A). We divided the ENP into location groups to account for larger geographical coverage and continued to observe significantly higher kinship in the GOC (Fig. S11B, C). In summary, these results reflect the greater historical isolation and small population size of the GOC[29] and a lack of recent inbreeding in both populations.

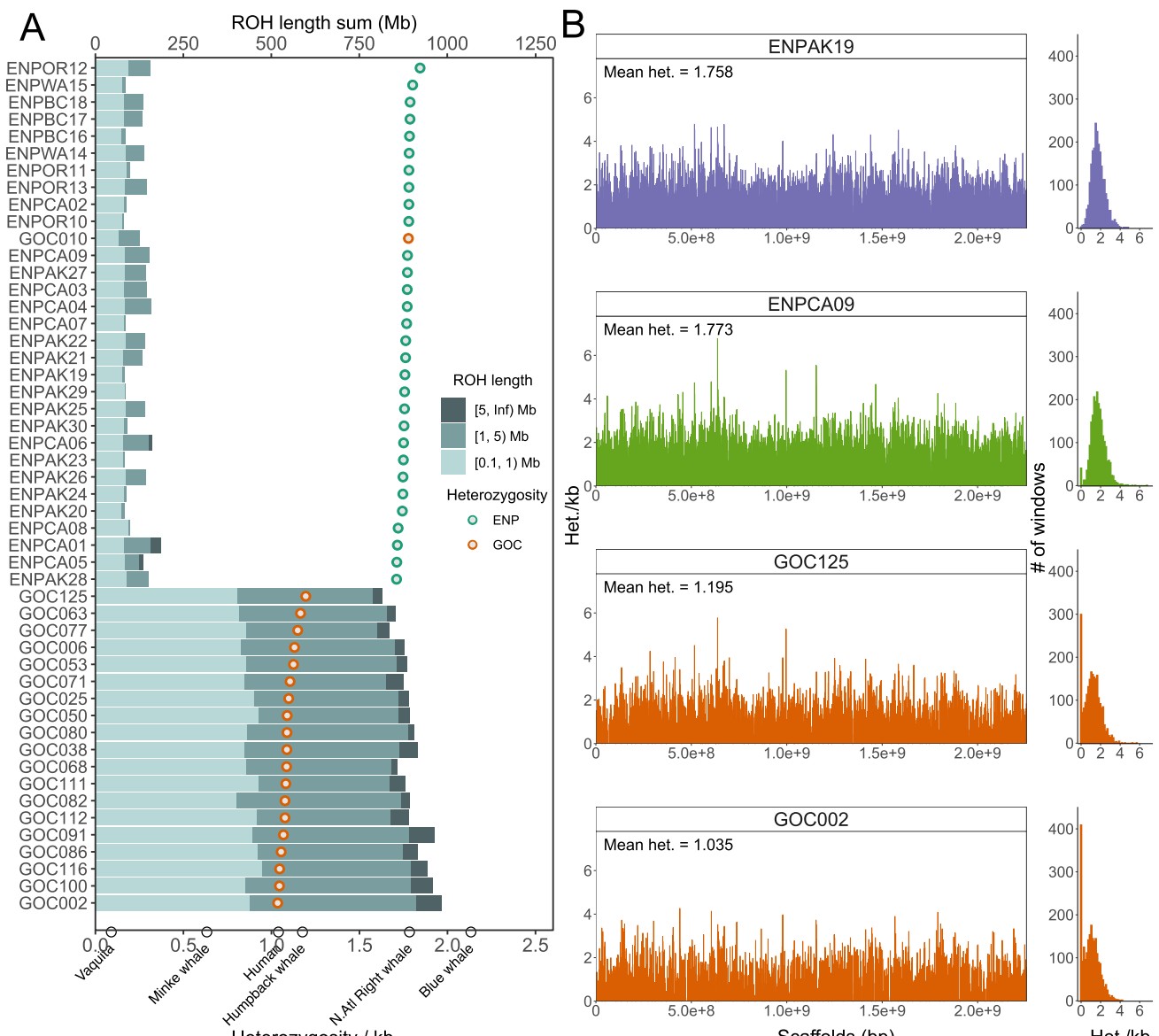

**Fig. 2 | ROH and distribution of heterozygosity across the genome. A** Points of genome-wide heterozygosity for each sample are ranked by decreasing heterozygosity from top to bottom. Circles at the bottom axis denote heterozygosity in other mammals. Barplots present summed lengths of short (0.1 Mb ≤ ROH < 1 Mb) to long (>5 Mb) ROH per individual (top axis). **B** The left panel shows per-site heterozygosity in non-overlapping 1-Mb windows across called scaffolds. The genome-wide heterozygosity value is annotated as "Mean het". The right panel summarizes the distribution of per-window heterozygosity. Individuals with divergent demographic histories were selected as examples. ENPAK19 represents the large outbred Eastern North Pacific population that recently experienced whaling. ENPCA09 is an admixed individual. GOC002 and GOC125 belong to the small, isolated Gulf of California population. Source data are provided as a Source Data file.

## Demographic inference of whaling, divergence and gene flow

We reconstructed the demographic history of fin whale populations using the site frequency spectrum (SFS) to assess the impact of whaling in the Eastern North Pacific population and to determine the demographic events that have shaped the genomic diversity of the Gulf of California population. First, using the SFS from each population, we tested different single-population effective size ($N_e$) change models, employing coalescent[43] (fastsimcoal2) and diffusion approximation[44] (∂a∂i) methods. We assumed a generation time of 25.9 years[45] and a mutation rate of 2.77E-08 mutation/bp/generation[37], and tested several nested models with increasing numbers of size-change epochs (Fig. S12). Both inference methods provided concordant findings and ∂a∂i results are shown throughout the text, except when noted (see Tables S5–S7, for fastsimcoal2 results and all 95% confidence interval [CI] values). Our demographic

analyses show that a 3-epoch model was the best fit for the ENP population (Figs. 3A, B, S13A; Tables S5, S6) and revealed an expansion starting ~115 thousand years ago (kya; 4,424 generations), from an ancestral $N_e$ of 16,479 to 23,913. This was followed by a severe decline only 26 (one generation ago for fastsimcoal2 estimate; 95% CI: 0–2) or 52 years before present (two generations ago for ∂a∂i estimate; 95% CI: 1.89–2.11) to a current $N_e$ = 305 individuals (95% CI: 0–1137; Fig. 3A, B; Table S7), representing an ~99% reduction. To further verify the timing and size of this recent population reduction, we implemented a grid search (Fig. S14, see Supplemental Methods and Supplemental Results), performed additional inference runs varying the time for the whaling reduction (Tables S5, S7), used different optimization methods (Table S8), confirmed our power to detect such recent decline using coalescent SFS simulations under this model (Fig. S15), and ran supplementary inferences under a SFS

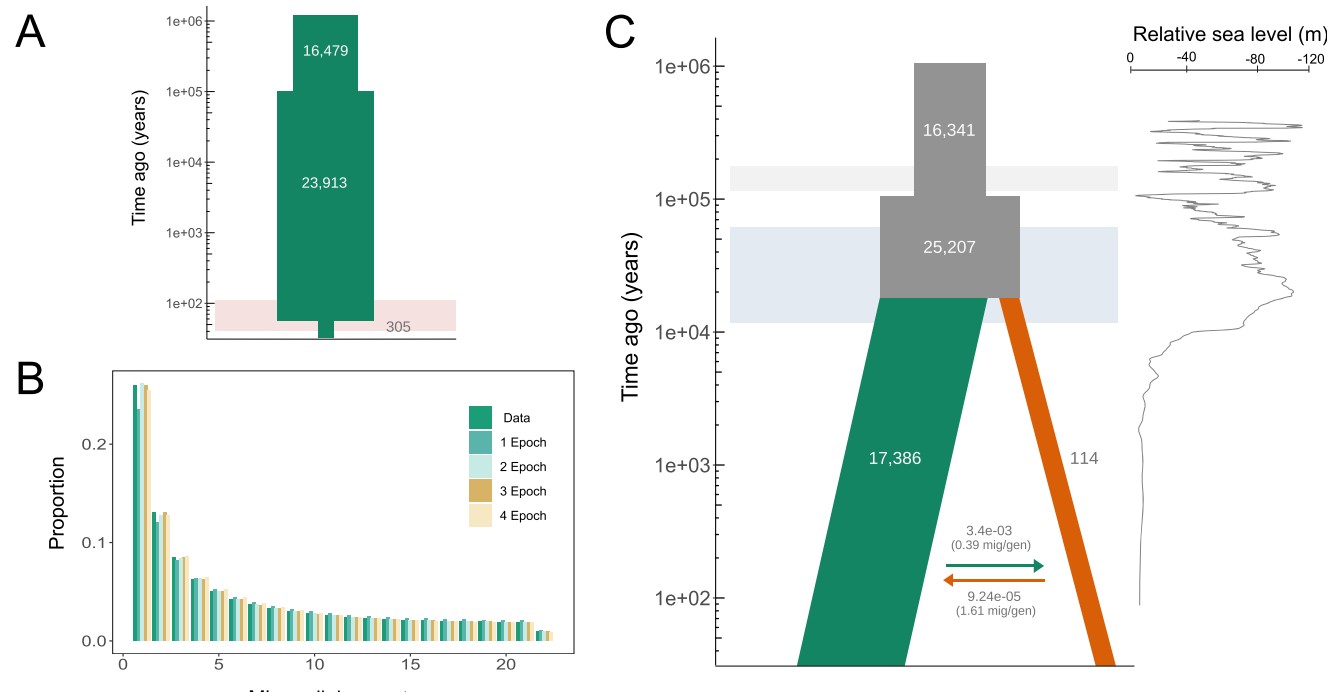

**Fig. 3 | Demographic history inferred for fin whale populations. A** The historical demography of the Eastern North Pacific (ENP; green) population is best represented by a single-population 3-epoch model. This model has an initial expansion, occurring around 115 thousand years ago (kya; 4424 generations) followed by an ~99% reduction only 26 to 52 years ago (one or two generations), during the whaling period for this species in the North Pacific (red horizontal bar). **B** Fit of the SFS from each demographic model (1- to 4-epoch) obtained with ∂a∂i for the ENP population to the SFS from the empirical data (Data). The SFS distribution for the 3-epoch model represented in A shows the best fit to the data. **C** Two-population model showing an ancestral effective population size expansion from approximately 16,000 to 25,000 individuals during the Eemian interglacial period

>100 kya (between the Illinois [gray bar] and Wisconsin [light blue bar] glaciations). The two populations diverged around 16 kya, during the Last Glacial Maximum. After the divergence, the ENP population (green) remained at an effective population size of ~17,000, whereas the Gulf of California (GOC; orange) population has remained small at an effective size of $N_e$ = 114. These populations have maintained low levels of asymmetrical gene flow, with higher migration rates from ENP into GOC (3.42E-03), than vice-versa (9.24E-05). However, when scaled by the receiving population's effective size, the GOC is only receiving 0.39 effective migrants/generation, while the ENP receives 1.61 effective migrants/gen. The black line to the right shows the relative sea level[114]. Source data are provided as a Source Data file.

without filtering on genotype calls to avoid bias against rare alleles (Tables S9, S10; Supplemental Methods and Supplemental Discussion). These additional analyses demonstrated that our findings reflect a drastic recent reduction one or two generations ago. Since the average collection year for samples from this population was 2006 (Table S1), the estimated times of the reduction correspond to the years 1954 to 1980, coinciding with the most intense whaling period this population suffered between 1940 and 1980[26,27].

For the Gulf of California population, none of the inferred SFS for the single-population models had a good fit to the data (Fig. S13B). Additionally, the models with the best likelihood did not show convergence or concordant parameter estimation between inference methods (Tables S5, S6, S7), which can indicate an over-parameterization of the models (see Supplemental Results). Therefore, we inferred the demographic history of the Gulf whales using a two-population model (described below) because they have shown to contain more information than single-population models and improve demographic inference[46].

The time of divergence and migration rates between both populations were estimated by testing several two-population models based on the joint SFS between ENP and GOC (Figs. S16, 17; Table S5). The model of an ancestral size change before the populations diverged fits our data well (Figs. 3C, S17; Table S5), is consistent among inference methods (Tables S11, S12) and is biologically feasible, therefore it was chosen as our best model (see Supplemental Results). This model predicted that before the populations separated, the ancestral population expanded from ~16,000 effective individuals to ~25,000, more than 100 kya (4322 generations). Then, the populations split between

16 and 25 kya (616 and 960 generations, ∂a∂i and fastsimcoal2 estimates, respectively). Thereafter, the ENP population remained at $N_e$ = 17,386 until it recently crashed due to whaling, as shown by the single-population model. By contrast, the GOC effective population size remained small after the divergence at $N_e$ = 114. The model also inferred asymmetrical gene flow, with a higher migration rate from the Pacific into the Gulf population (3.42E-03; fraction of individuals that are migrants) than in the opposite direction (9.24E-05; Table S11). However, when scaled by the receiving population's effective size, these rates represent a long-term effective migration of 0.39 immigrants per generation into the Gulf and 1.61 into the Pacific population (Fig. 3C).

To test if unsampled (ghost) populations contributed to migration into the GOC, we ran additional two-population models incorporating feasible ghost populations, the South Pacific and the western North Pacific (WNP). The ghost western North Pacific had a higher log-likelihood (Table S13) but did not considerably increase the total migration into the Gulf of California (the migration rate and effective migration from the ghost WNP into the GOC were 2.09E-04 and 0.01, respectively; Table S14; Fig. S18), demonstrating that migration from ghost populations into the GOC is negligible and does not affect our estimates. However, ghost population models revealed that the divergence between the ancestral ENP and ghost WNP populations match the expansion observed in both the single-population ENP and two-population models, around 4300 generations ago (Supplemental Discussion; Figs. 3A, C, S18; Tables S7, S11, S14).

Our results suggest the GOC population was founded at the end of the Wisconsin glaciation during the Last Glacial Maximum[47] and

remained small and highly isolated since then, receiving less than one migrant per generation (Fig. 3C). These findings are substantially different from estimates based on mitochondrial and microsatellite loci that predicted more recent divergence times, ~2300 or 9300 years before present (123 or 360 generations ago, respectively) and ~1 migrant per generation[30,31] (see Supplemental Discussion). Therefore, our results emphasize the greater resolution of whole-genome resequencing data for demographic inference empowered by the sheer availability of independent genealogies sampled[20] compared with only a handful of microsatellite loci[30] and a maternally inherited non-recombining marker.

## Putatively deleterious variation and genetic load

Our demographic inference analysis suggests a historically large population size and a recent contraction for the ENP population and a high degree of isolation for the GOC population. To assess how these demographic trajectories have impacted fitness, we examined variants in coding regions, which are more likely to have functional impacts. The derived alleles were classified into four mutation types: synonymous, tolerated nonsynonymous (SIFT score ≥0.05), putatively deleterious nonsynonymous (SIFT score <0.05), and loss-of-function (LOF; identified using snpEff, details in Methods). The synonymous and tolerated nonsynonymous mutations serve as a proxy for neutral variants whereas the putatively deleterious nonsynonymous and LOF mutations are proxies for putatively deleterious variants[48]. Although amino-acid changing variants could serve as candidates for local adaptation, most of them are deleterious[49,50]. Since the dominance for variants in natural populations is poorly quantified, we assumed two extreme scenarios. Specifically, the dominance of all variants is fully recessive ($h = 0$), or fully additive ($h = 0.5$).

For all four mutation types, heterozygosity is significantly depleted and homozygosity is significantly elevated in the GOC population (MWU tests $p = 2.9E\text{-}12$ in all comparisons; Table S15). This pattern has not been reported in other fin whale populations or great whale species[25] and is consistent with reduced genome-wide heterozygosity and small population size. The number of homozygous derived putatively deleterious nonsynonymous genotypes per individual was on average 39.68% higher in the GOC (2079) compared to the ENP population (1488). Similarly, the number of homozygous-derived LOF genotypes was on average 28.98% higher in the Gulf (140) compared with the Pacific population (108; Fig. 4A). Assuming that these putatively deleterious mutations are also at least partially recessive, this increased homozygosity in the GOC is predicted to result in reduced fitness[51].

When deleterious mutations act in an additive manner, the genetic load is determined by counts of derived alleles per genome. We found that the ENP and GOC populations showed a similar number of derived neutral alleles as expected[52] (Table S15). For the putatively deleterious class of mutations, only nonsynonymous alleles showed a significant 2.03% elevation in the GOC population (GOC average = 5983, ENP average = 5864, MWU test $p = 1.20E\text{-}07$), whereas the number of LOF alleles were similar in the two populations ($p = 0.87$; Fig. 4B). Assuming that these nonsynonymous alleles are slightly deleterious, the small population size of the GOC population likely increased the strength of genetic drift and decreased the efficacy of selection compared to the larger ENP population, allowing the persistence of deleterious variants in the Gulf. By contrast, the similar number of LOF alleles indicates that, in spite of the GOC population's small size, purifying selection has remained effective at eliminating the most deleterious mutations. Overall, these results imply a slight increase in the genetic load in the GOC population if deleterious mutations are additive.

Finally, we computed the $R_{XY}$ (relative accumulation of derived alleles) and $R^2_{XY}$ (relative accumulation of derived homozygotes) statistics that compare the expected number of the derived alleles or

homozygotes occurring only in one population[53] (Fig. 4C). Among the four mutation types, only the deleterious nonsynonymous alleles showed a relative accumulation of derived alleles in GOC ($R_{GOC/ENP} = 1.04$, Z-score $p = 0.02$), similar to the allele counts pattern (Fig. 4B). However, the $R^2_{XY}$ was significantly elevated for all mutation types in the GOC population (Z score $p < 0.001$ for all comparisons), consistent with their higher homozygosity values in GOC (Fig. 4A). We repeated these analyses using snpEff's mutation impact categories (i.e., high, moderate and low) to rule out software bias (see Methods) and found similar results (Fig. S19). In summary, these results suggest an increase in genetic load in the GOC population, both due to a shift towards higher homozygosity among all protein-coding variants, as well as an overall accumulation of putatively deleterious nonsynonymous alleles compared to the ENP population. However, the magnitude of the effect on fitness is unclear, given uncertainties about the selection and dominance coefficients of these mutations[51].

## Simulations of deleterious variation and genetic load

To further explore how fin whale demographic history and the recent whaling-induced decline has shaped patterns of deleterious variation and accumulation of genetic load, we ran forward-in-time genetic simulations using SLiM v.3.3.2[54]. We simulated a 10 Mb chromosomal segment with a combination of intergenic, intronic, and exonic regions. Selection coefficients for nonsynonymous deleterious mutations were drawn from a distribution estimated from humans[55], and dominance coefficients were set such that the most deleterious mutations were highly recessive, though nearly neutral mutations were closer to additive (see Methods for details).

Using this simulation framework, we first investigated the extent to which the recent whaling bottleneck may have led to an increase in genetic load in the ENP population. Specifically, we simulated under our best-fit ENP demographic model, which includes a contraction to $N_e = 305$ two generations ago (Fig. 3A). After two generations at $N_e = 305$, we did not observe any changes in genetic load, heterozygosity, or levels of inbreeding, as expected given the short duration of this decline (Fig. 5A). To explore how various potential recovery scenarios may impact the viability of the ENP population in the future, we continued these simulations for an additional 18 generations following the decline, during which we observed increasing trends for genetic load and levels of inbreeding, though minimal impacts on genetic diversity (Fig. 5A). To test the impacts of a partial recovery in the ENP, we also ran simulations where we increased the effective population size to $N_e = 1000$ after two generations at $N_e = 305$. Here, we observe minimal increases in genetic load and inbreeding, suggesting that even a modest recovery would stave off any deleterious genetic effects (Fig. 5A). In conclusion, these results highlight the importance of a prompt recovery to minimize deleterious genetic impacts from the whaling bottleneck.

Our next aim for these simulations was to assess the importance of low levels of migration (0.39 effective migrants/gen from ENP to GOC) for maintaining genetic diversity and fitness in the small GOC population ($N_e = 114$) despite long-term isolation (~16 kya). We simulated under our best-fit two-population demographic model, running simulations that included the estimated rates of migration between the ENP and GOC (Fig. 3C) as well as simulations where no migration was allowed. When carrying out simulations that include the empirically inferred rate of migration from ENP to GOC, we observe a 26.7% reduction in heterozygosity and increase in $F_{ROH > 1Mb}$ from 0 to 0.10 in the GOC population compared to the ENP population (Fig. 5B), in good agreement with the trends from our empirical dataset (35.7% empirical heterozygosity reduction; Fig. 2). Additionally, we find that average genetic load in the GOC population is elevated to 7.75% compared to 2.87% in the ENP population (Fig. 5B). However, this increase in genetic load appears to be counteracted by the removal of recessive strongly deleterious mutations ($s < -0.01$), which are reduced in frequency by

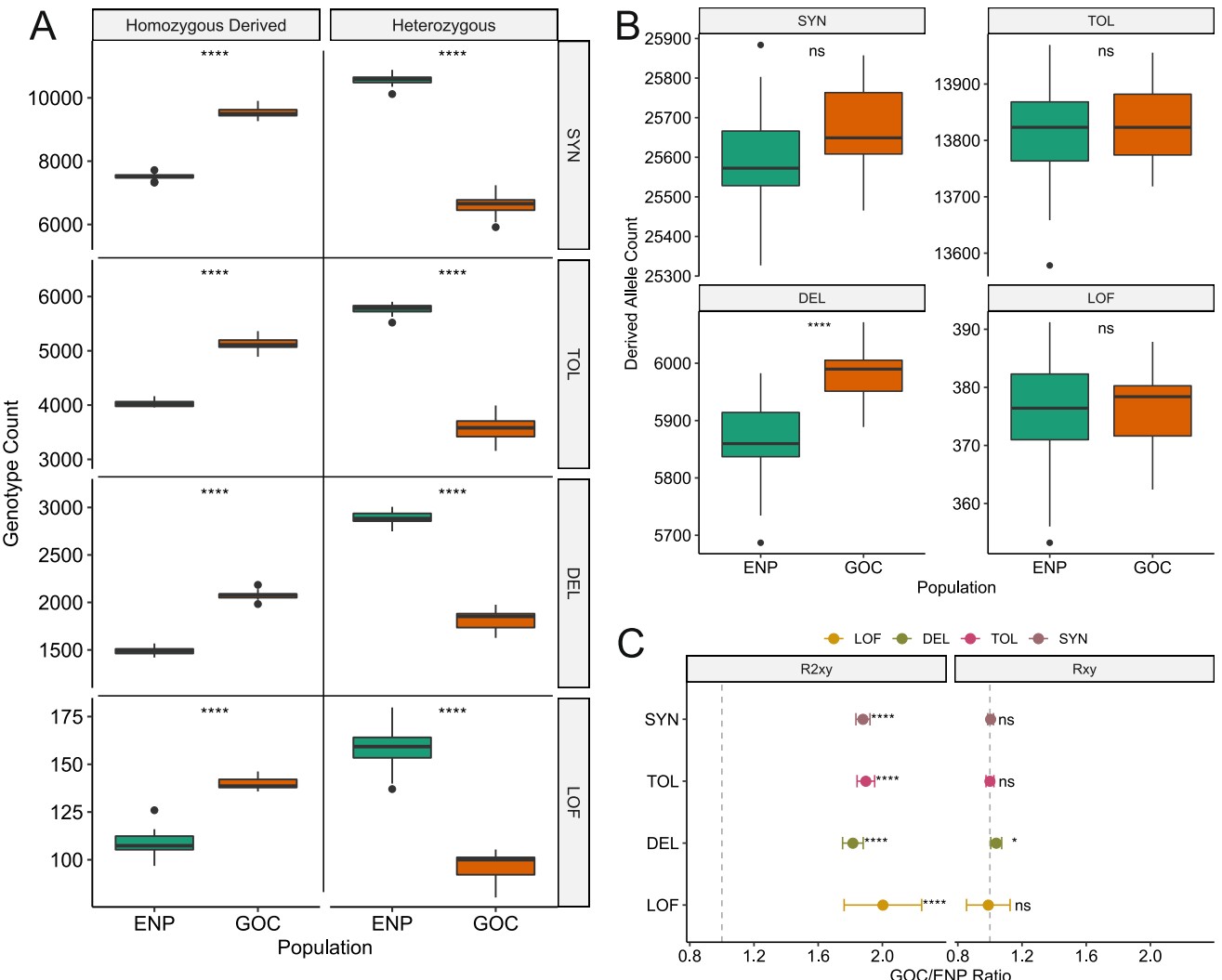

**Fig. 4 | Increase in putatively deleterious variation in the GOC compared to the ENP fin whales.** Sample sizes: Gulf of California (GOC) $N = 17$, Eastern North Pacific (ENP) $N = 27$. **A** The GOC fin whales contain significantly fewer heterozygous and more homozygous derived genotypes in all four functional categories of variants. **B** Only putatively deleterious nonsynonymous alleles (DEL) are significantly elevated (two-tailed MWU test $p < 0.001$; Table S15) in the GOC compared with the ENP population. The ENP and GOC fin whales contain similar numbers of derived neutral alleles (SYN: synonymous and TOL: tolerated nonsynonymous), and putatively deleterious loss-of-function (LOF) alleles. For **A** and **B**, we used two-tailed Mann-Whitney U tests without multiple testing adjustment (the exact $p$ values for the Mann–Whitney $U$ tests are given in Table S15 in the supplementary material). In the boxplots, the notch indicates the median, and the boxes represent

the 25th and 75th percentiles. The whiskers extend to data points no >1.5 * IQR (inter-quantile range) from the hinges and the points show outliers beyond the whiskers. **C** $R_{XY}$ and $R^2_{XY}$ statistics in GOC ($X$) and ENP ($Y$) populations. $R_{XY} > 1$ (dashed gray line) indicates a relative accumulation of the corresponding mutation category in the GOC population. Similarly, $R^2_{XY} > 1$ indicates relative accumulation of homozygous mutations. The 2x standard error based on the jackknife distribution is denoted as error bar, the circles in the center of the error bars represent the $R_{XY}$ or $R^2_{XY}$ values. For C we used a two-tailed $Z$ score test without multiple testing adjustment ($R_{XY}$ $Z$-test significant values: $p_{SYN} = 0.61$, $p_{DEL} = 0.02$, $p_{TOL} = 0.98$, $p_{LOF} = 0.88$; $R^2_{XY}$ $Z$-test significant values: $p_{SYN} = 0$, $p_{DEL} = 2.60e\text{-}142$, $p_{TOL} = 3.73e\text{-}234$, $p_{LOF} = 9.91e\text{-}17$). Significance levels: ns, not significant; *$p < 0.05$; **$p < 0.01$; ***$p < 0.001$; ****$p < 0.0001$. Source data are provided as a Source Data file.

22.9% in the GOC population (Fig. S20). By contrast, we observe minimal differences in the numbers of moderately ($-0.01 < s \leq -0.001$) or weakly ($-0.001 < s \leq -0.00001$) deleterious alleles per individual (Fig. S20), suggesting that migration has helped keep these mutations from drifting to high frequency in the GOC population. In summary, these results suggest that isolation and small population size in the GOC may have resulted in a lowered fitness, though these fitness reductions have apparently not been substantial enough to impact population viability.

When simulating without migration, we observed far more dramatic changes in the genetic composition of the GOC population. Specifically, we found a near-complete loss of genetic diversity, higher levels of inbreeding ($F_{ROH>1Mb} = 0.11$), and a substantial increase in genetic load to 10.3% in the GOC population (Fig. 5B). The loss of

diversity was also confirmed in theoretical calculations (see Supplemental Results). This increase in genetic load appears to be driven primarily by fixation of moderately deleterious alleles (9.22% gain in the isolated GOC population compared with the migration scenario; Fig. S20). Thus, these simulations suggest that, in the absence of migration, the GOC population would have experienced a much more substantial increase in genetic load, which may have been substantial enough to drive extinction. In conclusion, these results highlight the importance of low levels of migration in maintaining viability in the GOC population over its long period of isolation.

## Discussion

Detecting recent population bottlenecks in endangered species using estimates of genetic diversity in contemporary samples has been

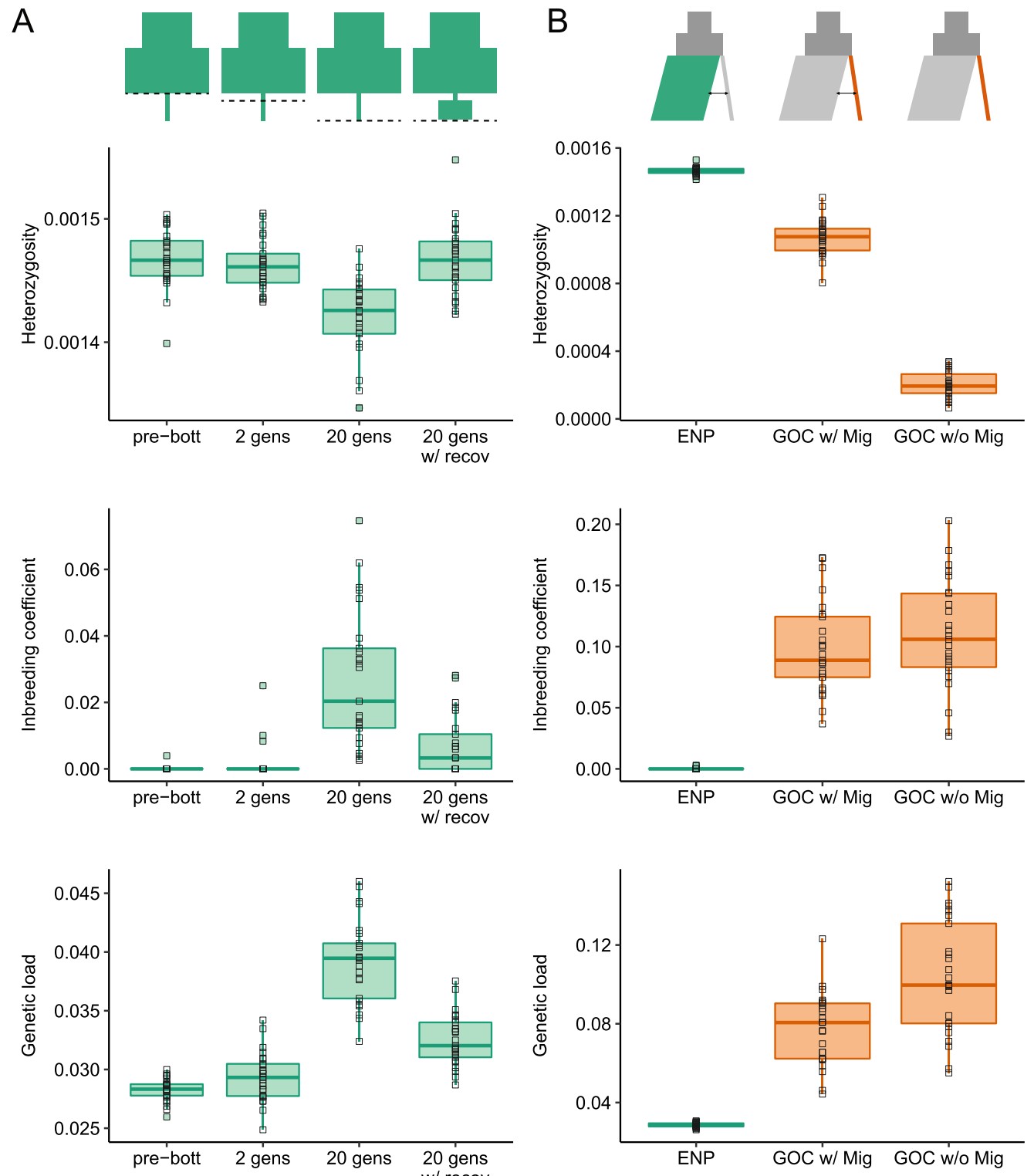

challenging[19,20], especially in long-lived species with long generation times, such as the great whales[21,56]. Specifically, the influence of changes in population size on genetic diversity is slow relative to temporal scale of human-induced events[19] and the overall loss of genetic variation depends on the duration of the bottleneck relative to the life history traits[57,58] such as life-span and generation time. Although genomic data can improve our ability to detect the impact of bottlenecks, most studies analyzing whole-genome data have failed to detect signals of whaling in blue[38] and gray whales[59], presumably due to small sample sizes. Recently, low-coverage sequencing of North Atlantic fin

whales may have recovered a signal of whaling, although the results did not completely rule out the alternative scenario of a more gradual decline over the last 600 years rather than an abrupt whaling bottleneck[25], two scenarios which are challenging to disentangle, particularly with added uncertainties associated with low-coverage data. Here, we show that using high-coverage genome resequencing (~27×), sampling a high number of individuals (~30 per population) at a single timepoint, and implementing SFS-based demographic inference approaches, anthropogenic population contractions, such as the one imposed by the 20th-century whaling on fin whales[26,27] can be

**Fig. 5 | Simulations of heterozygosity, inbreeding coefficient, and genetic load.** Representations of the demographic scenarios under which the simulations were performed are shown at the top. **A** Results for simulations under single-population 3-epoch model for the ENP population (green), including mean heterozygosity, levels of inbreeding ($F_{ROH>1Mb}$), and mean genetic load. Each quantity was measured prior to the onset of the whaling bottleneck (pre-bott), after two generations at the bottleneck $N_e = 305$ (2 gens), after 20 generations at the bottleneck $N_e = 305$ (20 gens), and 20 generations following the onset bottleneck where recovery to $N_e = 1000$ occurred after just two generations at $N_e = 305$ (20 gens w/ recov). In the demographic representations, the dashed line indicates the timing of sampling. **B** Results for simulations under our chosen two-population model. Each quantity is shown for the ENP (green) and GOC (orange; GOC w/mig) populations at the end of the simulation. We also simulated under a no migration demographic scenario for the GOC population (orange; GOC w/o mig). Note the much lower heterozygosity, higher inbreeding, and higher genetic load in the GOC population in the absence of migration. In the demographic representations, the sampled population, ENP or GOC, are shown in green or orange, respectively, and the presence/absence of migration indicated with the black arrows. For all boxplots, the notch indicates the median, and the boxes represent the 25th and 75th percentiles. The whiskers extend to data points no >1.5 * IQR (inter-quantile range) from the hinges and the solid squares show outliers beyond the whiskers. Hollow squares denote each simulation's value. Source data are provided as a Source Data file.

identified (Supplemental Discussion). In addition to our sampling and methodological approaches, the combination of a high pre-whaling genetic variation possessed by the fin whales in the Eastern North Pacific[30,31,34,60] together with an extreme reduction of two orders of magnitude, even if short, likely caused a deficit in low-frequency variants in present-day individuals that we were able to detect[20] (Fig. 3B). Therefore, our research demonstrates that even very recent human-driven population bottlenecks leave a detectable genomic footprint in the SFS derived from genome-wide data of contemporary individuals, and this signal can be used to identify the demographic and genetic effects of recent exploitation and model current and future impacts on populations.

Our study examines the natural experiment of whale populations that have experienced both natural and anthropogenic population bottlenecks, providing unique contrasts not available in single-population studies[25]. Despite a 99% decline in effective population size, the Eastern North Pacific fin whales have retained most of their pre-whaling genetic diversity (Figs. 2, 5A). They do not exhibit a substantial decrease in genome-wide heterozygosity nor an increase in inbreeding or genetic load (Figs. 2, 4 and 5A), similar to that found in a North Atlantic population[25]. Since genetic diversity declines exponentially with the number of generations passed from the contraction, this lagging impact on genetic diversity is likely a consequence of the long generation time of fin whales[45] (~25.9 yrs) relative to the duration of the whaling bottleneck (~70 years) and a partial recovery following the whaling moratorium beginning in 1985[32,58,61]. The contraction, although severe, only lasted for two generations (see Supplemental Results). However, other detrimental effects remain alarming. The reduction in 99% of pre-whaling effective size has likely had strong ecological consequences[15,18,62]. Additionally, if the ENP population does not completely recover and remains relatively small, it may experience a loss of adaptive potential to resist future climate change or disease[63]. Furthermore, this reduced effective population size in the ENP could also imperil the viability of the Gulf of California population by further diminishing or completely halting migration into this population, which our simulations have shown can accelerate the accumulation of deleterious load and loss of genetic diversity. These simulations allowed us to explore genomic consequences under various conservation scenarios (Fig. 5), an important perspective not yet adopted in other great whale genomic studies[25,38,59]. Both empirical and simulation findings show that continuing the current moratorium and enhancing population size remains essential for fin whale recovery and long-term persistence[17,26].

Regarding the Gulf of California fin whale population, our results show that immigration from ghost populations is negligible (see Supplemental Discussion) and as few as 0.39 migrants per generation have been sufficient to maintain genetic diversity and fitness in this population over ~16,000 years of isolation (Fig. 5B), which is consistent with other genetic and ecological studies describing the isolation of this population[28,30,34]. By contrast, when omitting migration from our simulations, we observe a near-complete loss of genetic diversity and a substantial increase in levels of inbreeding and genetic load (Fig. 5B). Thus, these results highlight the importance of gene flow for maintaining population viability over long evolutionary timescales[11,64], even when levels of migration are far lower than the classic rule of thumb of 'one migrant per generation'[10]. This rule has been widely applied in conservation, however, it is based on a neutral model that makes numerous simplifying assumptions and does not consider deleterious variation[12]. Here, we combine empirical observations with more realistic models including deleterious variation to demonstrate that small populations can be maintained by exceedingly low levels of migration, even when modest levels of genetic load may accumulate[65]. These results have important implications for conserving other small and isolated populations, where maintaining high levels of migration may not be feasible.

Population persistence in the GOC also appears to be enabled in part by eliminating strongly deleterious mutations, as has been shown in other small vertebrate populations[66,67] including marine mammals[36]. Specifically, our simulations suggest a 22.9% reduction in the frequency of these mutations in the GOC (Fig. S20) due to its long-term small population size, occurring despite the impact of gene flow continually reintroducing these mutations[13]. However, we were unable to detect this decrement in our empirical dataset, where we observed similar numbers of putatively deleterious LOF mutations in the GOC and ENP populations (Fig. 4). This discrepancy could be partially explained by LOF mutations being an imperfect proxy of strongly deleterious variation[68,69], as shown in empirical studies[48]. Although it could be argued that some genomic patterns of deleterious variation might reflect local adaptation in the GOC population, this explanation seems unlikely. For example, only drift would cause increased homozygosity in all mutation categories as observed, specifically, increased homozygosity in synonymous variants is not expected under a scenario of local adaptation (Fig. 4A, C). Moreover, local adaptive events occur more rarely than genetic drift and purifying selection that is constantly ongoing in natural populations[70].

Here, we have assessed the genomic impacts of both natural and anthropogenic bottlenecks on the second-largest mammal. We demonstrate that it is possible to confidently estimate the magnitude and timing of recent human-driven population bottlenecks, and to determine the key role that gene flow and potential purging of deleterious variants play in the persistence of small isolated populations by analyzing whole-genome resequencing data from contemporary samples together with individual-based simulations. From a conservation perspective, our findings expose the severity of whaling and indicate that it is necessary to reassess the recovery goals for the ENP fin whales and the regional threatened status of the GOC population, which may warrant specific conservation actions to maintain gene flow and avert additional impacts from climate change, mortality by entanglement[28] or microplastic contamination[71]. Therefore, our study contributes to fulfilling the overdue promise of genomics to conservation biology concerning the genetic effects of very recent population reductions caused by anthropogenic activities and identifying the evolutionary and ecological processes that promote the viability of small populations[72]. Finally, we demonstrate the importance of using both genomic and simulated data to inform the conservation of intensely exploited species.

## Methods

### Samples and sequencing

Tissue samples from 50 fin whales (*Balaenoptera physalus*) were collected using a standard protocol to obtain skin biopsies from free-ranging cetacean species, which use a small stainless-steel biopsy dart deployed from a crossbow or rifle[73,74]. These samples were collected throughout the Eastern North Pacific (ENP; $N = 30$, represented by individuals from the coasts of California [9], Oregon [4], Washington [2], British Columbia [3], and Alaska [12]; Table S1), and the Gulf of California (GOC; $N = 20$, from seven different localities; Bahía de La Paz [3], Loreto [6], Bahía de los Angeles [5], Bahía Kino [3], North of Tiburon Island [1], Puerto Refugio [1] and out of Bahía Los Frailes [1]). All samples from the Gulf of California were obtained under the appropriate collecting permits issued by the Mexican Wildlife Agency (Dirección General de Vida Silvestre, Subsecretaría de Gestión para la Protección Ambiental, Secretaría del Medio Ambiente y Recursos Naturales; permit numbers: D0070(2)−0598, D00700(2)−14093, D00750-1537 and SGPA/DGVS/−0576). Samples from the Eastern North Pacific were collected by the Southwest Fisheries Science Center (California, USA) under US Marine Mammal Protection Act permits (NMFS-873, NMFS-1026, NMFS-774-1437, NMFS 0782-1438, NMFS-774-1714, NMFS-774-1437, NMFS-14097, and NMFS-19091). DNA from the samples was extracted using the QIAamp DNA Mini Kit (Qiagen; California, USA. Catalog number: 51304). The genomic libraries were prepared from extracted DNA using the Illumina TruSeq DNA PCR-free standard kit (Illumina; California, USA. Catalog number: 20015962) following the manufacturer's instructions. Whole-genome sequencing was performed using the 150-bp paired-end protocol on Illumina HiSeqX or NovaSeq6000 platforms. Library preparation and sequencing were performed in Fulgent genetics' sequencing core facility (Fulgent genetics LLC; California, USA).

To compare the fin whales' genomic characteristics within Mysticeti, previously generated whole-genome resequencing fastq data from four representative Mysticeti species were downloaded from the NCBI Sequence Read Archive: the minke whale (*Balaenoptera acutorostrata*; SRR1802584), a stable and abundant rorqual; the humpback whale (*Megaptera novaeangliae*; SRR5665639), the closest relative with fin whales; the North Atlantic right whale (*Eubalaena glacialis*; SRR5665640) and the blue whale (*Balaenoptera musculus*; SRR5665644), the most endangered baleen whales (Table S1).

### Read processing and alignment

We followed the sequence reads processing and genotyping pipeline adapted from the Genome Analysis Toolkit (GATK) Best Practices Guide[75]. Read quality was first checked using FastQC v.0.11.8[76]. Illumina adapters were removed from the paired-end sequence reads using picard (v.2.20.3) MarkIlluminaAdapters. The adapter-free paired-end reads were aligned against the minke whale (*Balaenoptera acutorostrata scammoni*) reference genome (GCF_000493695.1 [BalAcu1.0]; Scaffold N50: 12,843,668, Downloaded on November 12, 2019) using BWA-MEM v.0.7.17[77]. Mapping statistics were generated using QUALIMAP v.2.2[78] and samtools v.1.9[79]. We used the minke whale genome as a reference because the available fin whale genome assemblies are much more fragmented and poorly annotated (GCA_008795845.1; Scaffold N50: 871,016) or they did not have a publicly available genome annotation as of November 2022 (GCA_023338255.1), and the blue whale genome (GCF_009873245.2) did not have genome annotation in 2019 (Supplemental Methods; Table S16; Fig. S21). The fin whale and minke whale are in the same genus, with a divergence time of ~10 million years ago[38]. The average mapping rate of fin whale reads to the minke whale genome is $99.09 \pm 0.21\%$ (Table S1), which is similar to the 99.49% mapping rate to the most recent fin whale reference genome (GCA_023338255.1; Table S2), obtained from a subset of samples ($n = 10$; see Supplemental Methods), suggesting that the divergence time with minke whales did not strongly impact read alignment.

### Genotype calling and filtration

Joint genotype calling at all sites (including invariant positions) across the reference genome was performed using GATK[80] (v.3.8). We removed PCR duplicates from the bam files using picard *MarkDuplicates*. Raw variant calling was performed for each individual using GATK's *HaplotypeCaller* using the default settings for removing low-quality reads (min_mapping_quality_score=20; min_base_quality_score=20). Joint genotype calls for the 50 fin whales were generated from the raw variants using GATK *GenotypeGVCF*, excluding scaffolds shorter than 1 Mbp. The total scaffold length used for genotyping was 2,324,429,847 bp, with the excluded scaffolds comprising only 4.4% of the total genome length (107,257,851 bp out of 2,431,687,698 bp).

Since we do not have a database of known variants, we did not perform base quality score recalibration (BQSR) or variant quality score recalibration (VQSR). Instead, we performed a stringent set of quality and depth filters for the genotype calls, keeping only high-quality biallelic SNPs and monomorphic sites with the latter including all homozygous reference or all homozygous alternate genotypes (Fig. S22). Sites that (1) had low Phred score (QUAL < 30); (2) failed GATK recommended hard filters (QD < 2.0 || FS > 60.0 || MQ < 40.0 || MQRankSum < −12.5 || ReadPosRankSum < −8.0 || SOR > 3.0); or (3) fell within repeat regions identified by WindowMasker[81], RepeatMasker[82] or CpG islands identified by UCSC genome browser (total length: 1,247,900,490 bp), were marked as failed filtration (Fig. S22A). For the sites that passed the above filters, we performed genotype-level filtration. Specifically, for each individual, only genotypes with a minimum depth of eight reads and maximum depth of 2.5x mean depth; a minimum Phred score of 20, and expected allele balance (the following thresholds were used for the allele balance, defined as the read depth for the reference allele divided by the total read depth: ≥0.9 for homozygous reference genotypes; between ≥0.2 and ≤0.8 for heterozygous genotypes; and ≤0.1 for homozygous alternative genotypes) were kept. Genotypes that failed these filters were converted to missing (Fig. S22B). Thereafter, sites were further filtered if they had more than 20% missing genotypes or more than 75% heterozygous genotypes (Fig. S22A). We repeated the genotype calling and filtration pipeline with four additional baleen whales included with 50 fin whale samples. The derived dataset ("f50b4" in the following text) was only used in the construction of neighbor-joining tree and generation of genome-wide heterozygosity comparison. An additional variant dataset ("genotype-filter-free" dataset) for the ENP individuals without any genotype-level filters was generated and used in confirmatory demographic inference (Supplemental Methods). We also performed the same genotyping pipeline using the most recent fin whale genome as reference (GCA_023338255.1) in a subset of 10 individuals (10-fin-ref dataset) to determine if there were significant differences in genomic diversity estimates caused by the reference genome used (minke whale vs fin whale; see Supplemental Methods, Results, and Discussion). The total number of sites that passed all the filters in our genotyping pipeline for the different datasets we analyzed is reported in Table S17.

### Variant annotations and identification of neutral regions

We annotated variant sites using two softwares, snpEff v.4.3.1[83] and SIFT4G v.6.0[84]. We used the minke whale genome annotation gtf file to build custom snpEff and SIFT4G databases with default settings. We then annotated and predicted the effects of variants with -canon option in snpEff and -t option in SIFT4G. The most deleterious effect was selected per site.

Although a recent fin whale genome assembly (GCA_023338255.1) has been annotated[25], this annotation is not publicly available at the present time, preventing us to use it to identify putatively neutral regions for our demographic and deleterious variation analyses. In addition, if the annotation of this fin whale genome assembly would be available it is unlikely it will significantly affect our main results and conclusions (See Supplemental Discussion).

We used the minke whale as an outgroup to classify the allele ancestral states, and considered the sites in the minke whale reference sequence as ancestral. Because the minke whale has evolved since the common ancestor with these two populations of fin whales, the ancestral alleles identified may not represent the true ancestral state. However, this error is not expected to bias the relative comparison of variants between the ENP and GOC fin whales since they are equally diverged from the minke whale. To detect the putatively neutral regions for demographic modeling, we first extracted sites that passed all filters and are at least at 20 kb distance from exons or coding regions and not in CpG islands or repetitive regions using bedtools v.2.28.0[85]. The identified regions were aligned to the zebra fish genome, using BLAST v.2.7.1[86], regions with a hit with e-value lower than 1E-10 were further removed, as they could represent conserved regions and not evolving neutrally. 397,627,899 sites were defined as neutral.

## Evaluation of population structure

Population structure analyses were performed using the R package SNPRelate v.1.16.0[87] and gdsfmt v.1.22.0[88]. We selected biallelic sites in the vcf that passed variant filtration criteria and converted them to gds format using function *snpgdsVCF2GDS*. Linkage disequilibrium pruning was implemented (*snpgdsLDpruning*) with an $r^2$ cutoff of 0.2, and a minor allele frequency cutoff of 0.10. A total of 30,350 SNPs were kept for PCA, kinship, and $F_{ST}$ analyses.

We performed the PCA analysis using the function *snpgdsPCA*. After observing the overall population structure, an additional PCA was performed within ENP individuals to inspect variation among locations. The kinship between sample pairs was assessed using PLINK's identity-by-descent method of moments approach (*snpgdsIBDMoM*). We calculated kinship at three different levels: (1) populations (groups: ENP and GOC), (2) sampling locations (groups: AK, BC, OR, WA, CA, and GOC); and (3) merged middle ENP locations combining samples from BC, WA and OR (groups: AK, MENP and GOC). The two-tailed MWU test was used to compare the average kinship coefficients among groups. $F_{ST}$ between populations, sampling locations and merged ENP locations were calculated using the Weir and Cockerham estimator[89], with a SNP missing rate at 20% (function *snpgdsFst*, missing.rate = 0.2). The significance of $F_{ST}$ was estimated using 999 permutations described in ref. 90. Due to the low sample size in BC, OR and WA locations, we only estimated the significance of $F_{ST}$ between populations and merged ENP locations. To determine the potential influence from population substructure within ENP on $N_e$ estimates, we calculated the population size inflation factor by $1/(1-F_{ST})$[33], using the highest $F_{ST}$ value found in the ENP.

The LD pruned SNP set was converted to PLINK ped format using function *seqGDS2VCF* in R package SeqArray v.1.26.2[88] and PLINK v.1.90[91]. ADMIXTURE[92] (v.1.3.0) analyses were performed using values of $K$ from two to six, with 10 iterations per $K$. Mean cross-validation (CV) error for each $K$ was used to select the best number of ancestral populations ($K$). To further test a substructure in the ENP, additional ADMIXTURE analyses were performed within ENP individuals, using values of $K$ from one to six, with the same settings described above. A neighbor-joining phylogenetic tree was constructed from 32,191 LD pruned SNPs in the "f50b4" dataset using function *nj* in R package ape v.5.3[93], and visualized using ggtree v.2.0.4[94]. 1000 bootstraps were performed, and the North Atlantic right whale ("EubGla01") was designated as the outgroup (Fig. S5).

## Heterozygosity and identification of runs of homozygosity

We defined heterozygosity as the number of heterozygous genotypes divided by the total number of called genotypes, including monomorphic sites, that passed variant filtration standards[48]. We first calculated the genome-wide heterozygosity for all scaffolds used for genotyping. Two-tailed MWU tests were used to evaluate if the genome-wide heterozygosity varied significantly between the ENP and

GOC populations. We also calculated the per-site heterozygosity in non-overlapping 1 Mb windows across the scaffolds. Windows with more than 80% missing data were excluded. The missing data in these windows derive from regions that failed site filtering criteria described above.

For identifying ROH, we first separated the vcf file for ENP and GOC individuals and reestimated allele frequencies within each population. ROH were identified using *bcftools roh -G30* in bcftools v.1.9[39]. Three individuals were excluded from bcftools ROH analyses to avoid biasing allele frequency estimations [ENPCA09 and GOC010 due to admixture proportion > 0.25; ENPOR12 due to low genotyping rate (Fig. S22)]. Additional ROH analysis was performed using R package RZooRoH v.0.2.3[40], which can classify ROH segments into different age classes. A model with ten classes (9 ROH and 1 non-ROH) and a successive rate of three was applied (*zoomodel, K = 10, base = 3*). A minor allele frequency cutoff of 0.05 was used but no individual was excluded. For both methods, ROH segments less than 100 kb were discarded. The rest of the segments were divided in three length categories, short (0.1 Mb ≤ ROH < 1 Mb), intermediate (1 Mb ≤ ROH < 5 Mb) and long (≥5 Mb). The concordance of the two methods was confirmed (Fig. S9) and the output from the RZooRoH analysis is shown in the main text. The proportion of genomes with ROH ($F_{ROH}$) was calculated as the total length of ROH passing a certain length threshold (e.g. ROH > 100 kb) within an individual divided by the total scaffold length used for genotyping (2,324,429,847 bp). We used the two-tailed MWU test to compare total number of ROH segments in all length categories obtained in the two populations.

To determine if the inbreeding observed in both fin whale populations were due to recent or older events, we estimated the average time at which two haplotypes would coalesce in each of the ROH categories (short, intermediate and long). The length of ROH associated with inbreeding ($L$) decreases due to recombination in each generation and follows an exponential distribution[95–97]. The mean length of ROH in the exponential distribution is $E[L] = 100/2tr$, where $E[L]$ is the mean ROH length (in Mb), the constant 100 represents large segments belonging to the common ancestor in cM, $t$ is the number of generations to the common ancestor and $r$ is the assumed constant recombination rate of 1 cM/1 Mb[42,98]. Therefore, we calculated on average how many generations ago two haplotypes shared a common ancestor in each of the ROH categories as $t = 100/2E[L]r$[42].

## Projected site frequency spectra

A vcf file comprising only putatively neutral SNPs was used to obtain the site frequency spectrum (SFS) within and between populations. To avoid introducing bias to our demographic inferences from known contributing factors, such as uneven read depth[99], admixture proportions[44] and highly related individuals[100], six individuals were discarded in SFS projection (Low genotype depth: "ENPOR12"; Admixture proportion > 0.25: "ENPCA01", "ENPCA09", "GOC010"; Kinship > 0.15: "GOC080", "GOC111"). To avoid uncertainties in ancestral state classifications, we computed a folded SFS. This SFS was calculated based on a hypergeometric projection implemented using easySFS v.0.0.1 (https://github.com/isaacovercast/easySFS), which minimizes the effects of missing genotypes[101] (https://dadi.readthedocs.io/en/latest/user-guide/manipulating-spectra/#projection). From this projection, an optimal number of haploid individuals with a maximized number of SNPs are identified and this number is then used to construct the folded SFS. Both the single-population SFS for each population (projected haploid size: ENP = 44, GOC = 30; projected number of SNPs: ENP = 3,410,730, GOC = 1,532,968) and the joint two-population SFS were generated (projected number of SNPs: ENP-GOC = 3,418,226). Thereafter, the count of monomorphic sites was calculated and incorporated as follows: for the single-population SFS, monomorphic sites in the neutral regions that were called in at least the number of haploid individuals in the projection were added to the 0-bin already calculated

by the projection. For the two-population SFS, monomorphic sites were computed by counting the number of monomorphic sites that were called in at least 44 haploid individuals in the ENP population and at least 30 haploid individuals in the GOC population. These sites were added to the previous 0-0-bin of the projection.

## Demographic history reconstruction

We utilized the projected neutral SFS generated above to reconstruct the demographic history of fin whales surveyed in this study using two methods: ∂a∂i[44] (v.2.2.1; Diffusion Approximations for Demographic Inference) and fastsimcoal2[43] (v.2.6; fast sequential Markov coalescent simulation).

To explore a variety of possible demographic scenarios, we first tested the following single-population models on the ENP and GOC populations separately (Fig. S12; Table S7). All the models are described forward in time. For population size parameters ($N_{ANC}$, $N_{CUR}$, etc.), all values are in units of numbers of diploids. For time parameters ($T$, $T_{CUR}$, etc.), all values are in units of generations. For the ENP population, we explored two additional 3Epoch models fixing the $T_{CUR}$ to two generations (3EpochTcur2) or three generations (3EpochTcur3).

1. 1Epoch: single epoch model with no population size change. This model provides a "null model" that estimates ancestral population size ($N_{ANC}$).
2. 2Epoch: two epoch model with one size change event, from the ancestral size ($N_{ANC}$) to the current size ($N_{CUR}$) occurring $T$ generations ago.
3. 3Epoch: three epoch model with two size change events. The first event changed from the ancestral size ($N_{ANC}$) to a bottleneck size ($N_{BOT}$) and lasted for $T_{BOT}$ generations. The second event changed from the bottleneck size ($N_{BOT}$) to the current size ($N_{CUR}$) occurring $T_{CUR}$ generations ago.
4. 4Epoch: four epoch model with three size change events. The first event changed from the ancestral size ($N_{ANC}$) to a bottleneck size ($N_{BOT}$) and lasted for $T_{BOT}$ generations. The second event changed from the bottleneck size ($N_{BOT}$) to a recovery size ($N_{REC}$) and lasted for $T_{REC}$ generations. The third event changed from the recovery size ($N_{REC}$) to the current size ($N_{CUR}$) occurring $T_{CUR}$ generations ago. For the 3Epoch and 4Epoch models, we note that despite the population sizes were named as a "bottleneck size" or "recovery size", we did not restrict the direction of size changes (expansion or contraction) for any events.

Next, we tested the following two-population models (Fig. S16; Table S11) to elucidate the divergence time and gene flow in the ENP and GOC populations:

1. Split-NoMigration: a simple population split model with no migrations. The ancestral population ($N_{ANC}$) diverged into the ENP ($N_{ENP}$) and GOC ($N_{GOC}$) populations occurring $T$ generations ago. Two populations remained isolated since then.
2. Split-SymmetricMigration: an isolation-migration model. The ancestral population ($N_{ANC}$) diverged into the ENP ($N_{ENP}$) and GOC ($N_{GOC}$) populations occurring $T$ generations ago. The ENP and GOC populations maintained a symmetric migration rate of $m$.
3. Split-AsymmetricMigration: another isolation-migration model. This model is similar to model 2 (Split-SymmetricMigration), but the ENP and GOC populations were allowed to have different values of migration rate, with $m_{ENP->GOC}$ measured as the fraction of individuals each generation in the GOC population that are new migrants from ENP, and vice versa for $m_{GOC->ENP}$.
4. Split-AsymmetricMigration-ENPChangeTw2: this model is based on model 3 (Split-AsymmetricMigration), but an ENP population size change event to $N_{ENP2}$ is introduced after population divergence, with a fixed $T_W = 2$ generations before present. This size

change event after divergence is used to model the impact of whaling bottleneck.
5. AncestralSizeChange-Split-AsymmetricMigration: this model is based on model 3 (Split-AsymmetricMigration), but an ancestral size change event from $N_{ANC}$ to $N_{ANC2}$ that lasted for $T_A$ generations was introduced before population divergence.
6. AncestralSizeChange-Split-Isolation-AsymmetricMigration: this model is based on model 5 (AncestralSizeChange-Split-AsymmetricMigration), but after population divergence, an isolation period lasted for $T_D$, during which there is no migration between the ENP and GOC populations. Asymmetric migrations between two populations occurred $T_C$ generations before present.
7. AncestralSizeChange-Split-AsymmetricMigration-GOCChange: this model is based on model 5 (AncestralSizeChange-Split-AsymmetricMigration), but after population divergence, the GOC population remained at $N_{GOC}$ for $T_D$ generations. The GOC population then experienced a size change event from $N_{GOC}$ to $N_{GOC2}$ that occurred $T_C$ generations before present.

To evaluate if unsampled (ghost) populations contribute to the total migration into the GOC population, we included two feasible ghost populations into the selected two-population model, the South Pacific (SP), which diverged from the North Pacific ~1.8 Mya according to mtDNA data[31]; and the Western North Pacific (WNP) population, which has been suggested to breed separately from the ENP[27] potentially since the recent Pleistocene's interglacial periods[23]. For our demographic inference with ∂a∂i, we ran only one ghost model using the same initial parameters as in our chosen model. The initial parameter for the divergence time of ghost population was set at the expansion time in the ENP population 3Epoch model, and the size of the ghost population was fixed to the size of the ancestral population before divergence to find the best parameter space. In contrast, for fastsimcoal2 we constrained the lower and upper bounds for the divergence time of the ghost populations based on the previous knowledge mentioned above to 35,000 - 200,000 generations ago for the SP population and 100 - 10,000 generations ago for the WNP. We also fixed the size of the ghost populations to 30,000 haploids, approximately the same size of the ancestral population before the divergence.

## Fastsimcoal

The coalescent simulation approach fastsimcoal2 was employed to infer parameters and composite likelihoods for the demographic models specified above. Each inference was performed using the Expectation-Conditional Maximization (ECM) algorithm[102], using 60 ECM cycles (-L 60), in which each E-step consisted of 1,000,000 coalescent trees (-n 1000000), computing only the SFS for the minor allele (-m) with the following command line.

fsc26 -t $header.tpl -e $header.est -n 1000000 -m -M -L 60 -q

The starting parameters were chosen from a uniform distribution with an imposed minimum value and flexible upper boundary. The expected SFS under the fastsimcoal2 model parameters were compared to the empirical SFS and the multinomial log-likelihood was calculated. For single-population and joint populations models, we performed 100 and 50 replicates of the inference, respectively, to confirm that both parameters and log-likelihoods converged and parameters with the maximum log-likelihood were chosen. This difference in the number of replicates is due to the inference of two-population model parameters being more computationally expensive and time-consuming. All estimated size parameters were obtained as the number of haploids and converted to diploids, whereas time parameters were inferred as the number of generations before present day. To control for inflations in log-likelihood estimates in models with more parameters, we performed a likelihood ratio test (LRT) for nested models with its more immediate complex model (e.g., 2Epoch vs.

1Epoch, 3Epoch vs. 2Epoch) using the equation: *–2 * [loglikelihood (simple)–loglikelihood (complex)]*. The LRT significance was evaluated with a chi-square test (i$\chi^2$) with one or two degrees of freedom, depending on the number of parameter differences between models.

The parameter confidence intervals were obtained using a parametric bootstrap[43] following the simulation functionality described in fastsimcoal2's manual (http://cmpg.unibe.ch/software/fastsimcoal26/man/fastsimcoal26.pdf page. 56). For each model, we simulated 100 SNP-based SFS from the best-fit parameters in the observed data with ~4 million (3,927,079 for ENP single-population models, 3,908,444 for GOC single-population models and 3,864,185 for two-population models) non-recombining segments of 100 bp, mimicking the same number of observed sites. Parameters were estimated from 20 random starting conditions for the 100 bootstrapped SFS datasets using the same settings as described above for the empirical data. 95% confidence intervals of the best-fit parameters were obtained adding and subtracting two standard deviations of the 100 bootstrap estimated parameters from the empirical best-fit parameters.

### ∂a∂i

For demographic inference using ∂a∂i, haploid sample sizes plus 5, 15, and 25 were used as extrapolation grid points[44]. Lower and upper bounds of model parameters were imposed based on prior knowledge of population history, and starting parameters under these boundaries were chosen from previous knowledge or outputs from nested runs and randomized with a fold=1. We used the *optimize_log* function as our optimization algorithm, and calculated the multinomial log-likelihood for the expected SFS obtained from each optimization.

Best-fit parameter sets of each model were scaled using $N_{ANC}$ calculated by the equation $\theta = 4N_{ANC}\mu L$, where $L$ is the total sequence length of the neutral region (392,707,916 bp for ENP single-population models, 390,844,414 bp for GOC single-population models and 386,418,461 bp for two-population models), $\mu$ is the fin whale mutation rate (2.77E-08 mutations/generation/bp)[37], and $\theta$ is the optimal value of theta for the given model. Population size parameters were adjusted by $N_{ANC}$ into diploids and time parameters were re-scaled by $2N_{ANC}$ into generations. The model uncertainty was assessed by estimating 95% confidence intervals of the best-fit parameters using a Godambe Information Matrix (GIM) with bootstrapped data[103]. The bootstrapped data was obtained by dividing the genome into fragments of 4 Mb and generating 100 bootstrap pseudo-replicate datasets by resampling from those, which in total amounts for sampling 400 Mb that approximates the length of the putatively neutral data analyzed in our demographic inferences.

One hundred replicates of each model were performed with randomized starting parameters to assess convergence of the inferred parameters and composite likelihood. Parameters with the maximum log-likelihood among replicates from each model were selected and the expected SFS under these parameters was compared with the empirical SFS. LRT was calculated as previously described.

Additionally, to ensure that the results from the ENP population 3-epoch model were in fact reflecting the recent bottleneck caused by whaling, we simulated the SFS under ∂a∂i's inferred demographic scenario using msprime v.0.7.4[104]. The simulated SFS were generated using a recombination rate of 1E-8 cross-over events per base pair per generation and a mutation rate of 2.77E-8 per base pair per generation[37], with 1000 replicates and a chunk size of 2 Mb. Visual inspection was performed to validate the fit of simulated SFS to the empirical data. We also performed ∂a∂i inference on msprime simulated SFS using the same settings for empirical SFS and tested if we could obtain similar parameter estimates as the empirical data to confirm that we had the power to detect a recent population contraction.

To account for the correlations of current population size ($N_{CUR}$) and time of most recent contraction ($T_{CUR}$), we carried out grid searches to find the range of possible parameter pairs that are within two log-likelihood units of the maximum likelihood estimate (M.L.E; see Supplemental Methods).

### Model selection

We selected the models that more likely represent the demographic history of the populations from the demographic models without any constraints (i.e., not fixing any of the parameters to a certain value). To select the best demographic model, we considered several features of our demographic inference results. First, the log-likelihood of the models should be the highest given the satisfaction of the following criteria. Second, a good fit of the expected SFS to the empirical SFS. Third, the estimated parameter values between the two inference methods that we used (i.e., fastsimcoal2 and ∂a∂i) should be consistent, especially the direction of population size change (expansion vs contraction). Fourth, the log-likelihood of the top 10 replicated runs for each model should converge. We consider that a model has good convergence if the log-likelihood difference between the best run and the 10th best run of the model was no more than 25 log-likelihood units. Fifth, the model should have significantly better LRT than the more immediate nested model and this LRT significance should be consistent in fastsimcoal2 and ∂a∂i. Sixth, the range of the confidence intervals should not be unrealistically large. Models meeting the above criteria, were chosen as the ones representing the demographic history of fin whale populations. For the ENP single-population model, after choosing the 3Epoch model according to the previous criteria, we tried to confirm the findings of this unconstrained model by running it with the parameter reflecting the time of the putative whaling bottleneck fixed at 2 and 3 generations. Results show that models with fixed parameters have better log-likelihoods and do not significantly change the parameter values obtained with the unconstraint model, indicating that the estimates of the unconstrained model are a good representation of the demographic history of this population. For the two-population models, we ran the Split-AsymmetricMigration-ENPChangeTw2 model with the time of the whaling bottleneck fixed at two generations, such model was not selected.

### Quantifying putatively deleterious variation

Two lines of evidence were used to quantify relative levels of putatively deleterious variation in the ENP and GOC populations. We focused on mutations within protein-coding regions, which are more likely to have direct fitness impacts and identified derived alleles within four mutation types: synonymous, tolerated nonsynonymous, deleterious nonsynonymous, and LOF. The nonsynonymous mutations were classified as putatively tolerated (SIFT score ≥0.05) or deleterious (SIFT score <0.05) based on phylogenetic constraints using SIFT4G[84]. The LOF mutations are predicted to eliminate or severely inhibit gene function and include splice acceptor, splice donor, start lost and stop gained mutations. LOF mutations were identified using the default settings in snpEff[83], which utilized the LOF definition in ref. 69. We normalized for differences in missing data across individuals by the average number of called genotypes using R package vcfR v.1.12.0[105]. Since the dominance for variants in natural populations is poorly quantified, we assumed two extreme scenarios: (1) when the dominance of all variants is recessive ($h = 0$) and the fitness is only reduced in homozygous derived genotypes; or (2) when variants are additive ($h = 0.5$) and the fitness decreases linearly to the number of derived alleles. The real-life fitness impact probably lies between these two scenarios. We did not assume dominant variants ($0.5 < h \leq 1$) given that segregating deleterious variants are very unlikely to be dominant[51].

First, two-tailed MWU tests were used to evaluate if the normalized count of derived alleles and homozygotes varied significantly between the ENP and GOC populations in these four mutation types[48].

The count of derived putatively deleterious alleles, including the deleterious nonsynonymous and LOF alleles, are considered a proxy for additive genetic load, while the count of derived homozygotes provides a proxy for recessive load[106,107].

Second, we calculated the relative accumulation of mutations $R_{XY}$ and homozygous mutations $R^2_{XY}$ for the four mutation types using methods adapted from ref. 53. Here we designated the GOC population as population $X$ and the ENP population as population $Y$. At each polymorphic site $i$, we defined $d^i_X$ as the count of derived alleles at that site in a sample of $n^i_X$ haploid genomes from population $X$ and $d^i_Y$ as the count of derived alleles in a sample of $n^i_Y$ haploid genomes from population $Y$. The expected number of derived mutations observed only in population $X$ but not in population $Y$ is defined as:

$$L_{X,notY} = \sum_i (d^i_X/n^i_X)(1 - d^i_Y/n^i_Y) \qquad (I)$$

And the expected number of homozygous derived mutations observed only in $X$ but not in $Y$ is defined as:

$$L^2_{X,notY} = \sum_i \left(1 - \frac{2d^i_X(n^i_X - d^i_X)}{n^i_X(n^i_X - 1)}\right)\left(\frac{2d^i_Y(n^i_Y - d^i_Y)}{n^i_Y(n^i_Y - 1)}\right) \qquad (II)$$

The ratio statistics is further defined as:

$$R_{XY} = L_{X,notY}/L_{Y,notX} \qquad (III)$$

$$R^2_{XY} = L^2_{X,notY}/L^2_{Y,notX} \qquad (IV)$$

The standard errors of $R_{XY}$ and $R^2_{XY}$ were estimated from a weighted-block jackknife[53]. If selection has been equally effective and mutation rates remain the same in both populations, the $R_{XY}$ and $R^2_{XY}$ statistics are expected to be 1. $Z$ score test was used to evaluate the significance of the deviation from the null expectation.

Lastly, we assessed the robustness of the four mutation types across the genome using an additional mutation impact scoring system implemented by snpEff. SnpEff classifies variants' impact severity into HIGH, MODERATE, LOW and MODIFIER categories based on their effect types. We excluded the MODIFIER category because these mutations are mostly non-protein-coding. We additionally limited the MODERATE and LOW categories within the gtf identified coding sequence (CDS) region to exclude non-protein-coding mutations as well. Two-tailed MWU tests and $R_{XY}$ analyses were performed as described above to evaluate the variation in the count of derived alleles and homozygotes (Fig. S19). For all above analyses, we removed the six individuals that were also discarded in the demographic inference.

### Genetic load simulations

We conducted forward-in-time population genetic simulations using SLiM v.3.3.2[54]. For our simulations, we assumed a 10 Mb chromosomal segment with a uniform recombination rate of 1E-8 cross-over events per base pair per generation and randomly generated intergenic, intronic, and exonic regions, following ref. 108. The length of the 10 Mb chromosomal segment was chosen as a tradeoff between computation efficiency and genomic representation. Within this chromosomal segment, mutations occurred at a rate of 2.77E-8 per base pair per generation[37], with deleterious (nonsynonymous) mutations occurring only in exonic regions at a ratio of 2.31:1 to neutral (synonymous) mutations[109]. Selection coefficients for deleterious mutations were drawn from a distribution estimated from human data[55]. We assumed an inverse relationship between selection coefficients and dominance coefficients, given empirical evidence that

strongly deleterious mutations also tend to be highly recessive[51,110]. Specifically, we assumed that strongly deleterious mutations ($s < -0.01$) were fully recessive ($h = 0.0$), moderately deleterious mutations ($-0.01 \leq s < -0.001$) were partially recessive ($h = 0.1$), and weakly deleterious mutations ($-0.001 < s \leq -0.00001$) were nearly additive ($h = 0.4$).

Using this simulation framework, we simulated under our two best-fit demographic models, including a single-population model for the ENP population, and a two-population divergence model for the ENP and GOC populations (see above for details). For both models, we assumed a burn-in duration of 10x the ancestral population size. During the simulation, we kept track of several quantities for each simulated population, including mean genetic load (the reduction in individual fitness, calculated multiplicatively across sites), mean genome-wide heterozygosity, mean inbreeding coefficient (here measured as $F_{ROH}$, where the minimum ROH length was 1 Mb), and the mean number of strongly deleterious alleles ($s < -0.01$), moderately deleterious alleles ($-0.01 \leq s < -0.001$), and weakly deleterious alleles ($-0.001 < s \leq -0.00001$) per individual. These quantities were estimated using a sample size of 40 individuals. For all simulations, we ran 25 replicates and averaged these quantities across replicates.

### Reporting summary

Further information on research design is available in the Nature Portfolio Reporting Summary linked to this article.

### Data availability

The raw sequence data generated in this study are deposited in NCBI's Sequence Read Archive (SRA) database under accession numbers SRR23615109 - SRR23615158 (BioSample SAMN33439338 - SAMN33439387; BioProject PRJNA938516; see Table S1 for details). The sequence data for the additional mysticete species used in this study are available in NCBI's SRA database under accession numbers SRR5665640, SRR1802584, SRR5665644, and SRR5665639, please see Table S1 for details. The cpg island data are available in the UCSC genome browser (http://hgdownload.soe.ucsc.edu/goldenPath/balAcu1/database/). The balenopterid genomes assemblies used for the comparison shown in Table S16 are available in NCBI's Assembly database under accession numbers GCA_008795845.1, GCA_023338255.1, GCF_000493695.1, GCF_009873245.2, GCA_004329385.1, or in the DNA Zoo database under accession names Balaenoptera_physalus (https://dnazoo.s3.wasabisys.com/index.html?prefix=Balaenoptera_physalus/) and Balaenoptera_ricei (https://dnazoo.s3.wasabisys.com/index.html?prefix=Balaenoptera_ricei/). Source data are provided in this paper.

### Code availability

The scripts used to perform the sequence data processing and analyses are publicly available in a GitHub repository that can be accessed through Zenodo[111] at https://doi.org/10.5281/zenodo.7980107.

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

## Acknowledgements

We dedicate this work to Robert K. Wayne, a pioneer in the field of conservation genetics and conservation biology. We would like to thank the members of the research program of marine mammals from Universidad Autónoma de Baja California Sur for their help collecting the samples in the Gulf of California. Sergio Flores-Ramírez for his initial support in the Gulf of California project. Cei Abreu-Goodger for his support and for providing laboratory space during the initial analysis of the data, Phil Morin for reviewing an early draft of the manuscript. Unpublished genome assemblies and sequencing data for the Rice's whale and fin whale are used with permission from the DNA Zoo Consortium (dnazoo.org). For the fin whale DNAzoo assemblies, the sample for the assembly was collected by The Marine Mammal Center under the Marine Mammal Health and Stranding Program (MMHSPR) Permit No. 18786-04 issued by the National Marine Fisheries Service (NMFS) in accordance with the Marine Mammal Protection Act (MMPA) and Endangered Species Act (ESA). The work at DNA Zoo was performed under Marine Mammal Health and Stranding Response Program (MMHSRP) Permit No. 18786-03. This work used computational and storage services associated with the Hoffman2 Shared Cluster provided by UCLA Institute for Digital Research and Education's Research Technology Group. This work was supported by the Mexican National Council for Science and Technology (CONACYT) grant FONCICYT/50/2016, National Science Foundation (DEB Small Grant #1556705), UCMEXUS-CONACYT collaborative grant 2006. SNM was supported by CONACYT Postdoctoral Fellowship 724094 and the Mexican Secretariat of Agriculture and Rural Development Postdoctoral Fellowship. M.L. was supported by the University of California, Los Angeles Department of Ecology and Evolutionary Biology (EEB) Summer Research Fellowship. K.E.L. and C.K. were supported by NIH grant R35GM119856 to K.E.L. A.C.B. was supported by the Biological Mechanisms of Healthy Aging Training Program NIH T32AG066574. M.J.P.-A. was supported by ANID under Grant Program FONDECYT Iniciación 11170182. E.P. and M.J.P.-A. were supported by ANID Millennium Science Initiative Program ICN2021_002.

## Author contributions

S.N.-M., A.M.-E., and R.K.W. conceived the study. A.M.-E. and R.K.W. contributed reagents, materials, and analysis tools. J.U.R., L.V.-G., and F.I.A. collected and contributed the samples and sample information. S.N.-M. carried out laboratory work. S.N.-M., M.L., P.N.-V., and C.K. performed the analysis of the data. A.C.B., J.A.R. and A.R. provided scripts for some analyses. A.M.-E., A.C.B., J.A.R., A.R., M.J.P.-A., E.P., and K.E.L. provided guidance and advised the project. A.M.-E., K.E.L., and R.K.W. performed funding acquisition. S.N.-M., M.L., P.N.-V., CC.K., and R.K.W. wrote the manuscript with input from all the authors.

## Competing interests

The authors declare no competing interests.
