## [Peer Review File · Nature Communications]

The genomic footprint of whaling and isolation in fin whale populationsEditorial Note: This manuscript has been previously reviewed at another journal that is not operating a transparent peer review scheme. This document only contains reviewer comments and rebuttal letters for versions considered at *Nature Communications*.

REVIEWER COMMENTS

Reviewer #1 (Remarks to the Author):

The authors have addressed all my comments including the performance of new analyses. I remain with three comments:

I find the claim in the discussion for purging to have enabled population persistence still a bit strong given that this is purely based on simulations of 0.4% of the genome (10 Mb, with lots of assumptions).

I am not sure to understand, why LOF should not be nonsynonymous mutations and hence there should be an overlap between the LOF and nonsynonymous deleterious mutations? In my eyes, a stop-gained mutation is a clear example of a nonsynonymous mutation.

Regarding the formula $L = 100/2tr$:

I still don't understand, why this is not exponential. In my eyes this function should say $L = 100/(2r)^t$, because at each generation the length is halved?

And a comment only regarding the author responses to one of my comments (I am fine with them not reanalyzing using genotype likelihood approaches):

I've never read that 20x coverage (still not very high and errors are still quite likely) could be too high for analyses using genotype-likelihoods. Other than the analyses potentially running slower, I don't foresee a problem.

Reviewer #2 (Remarks to the Author):

The authors use high coverage whole genome sequencing data on 50 individuals from two populations to unveil their demographic history considering their historical levels of whaling (non harvested population vs harvested population). This is the first time I had the opportunity to review this interesting paper, as I did not participate in the previous round of revisions. I found the paper very interesting and uses standard, albeit cutting edge, population genomic analyses to inform management and conservation of these two Fin whale populations. I have also checked the comments and answers of the prior two Referees and I mostly agree with their observations. I am also generally happy in how the authors have addressed the points raised by the prior Referees. However, in my opinion there are still some issues that needs to be solved that probably will need a major reanalysis of the data. Find below my detailed comments.

Detailed comments:

*I agree with Referee#2, who mentioned that the lack of a reference genome may be an issue, especially when aiming at high impact journal. It is true that the use a reference genome, that is not of the target species, is commonly used in intraspecific population genomic studies, although its potential biases and assumptions are rarely discussed (synteny needs to be assumed, intraspecific polymorphism may be underestimated due to the difference between the intraspecific divergence and the divergence to the reference, definition of derived vs reference alleles). I have checked the reference genomes available and I have found that a reference genome of the target species has

been recently published (Wolf et al. 2022), as well as another set of whole genome sequencing of 51 individuals from North Atlantic populations (in a paper that is relatively similar to this one but in another region). I was very surprised to find this publication in your reference list (ref 25) but not the reference genome of this publication in your comparisons looking on what genome to use (e.g. Fig S20 or Table S15), despite its apparent high quality (e.g. BUSCO Cetartiodactyla C:83.4%[S:82.1%,D:1.3%] F:4.1%,M:12.5%, N50 = 2,49xE07, L50=27, 1.7xE07 genes). I would strongly advise to reanalyse your data with this new reference genome, instead of using the genome of a species that diverged between 11.5 and 17.6 MYA. You may also consider to include the data of these populations (despite being ~10 fold), as they have been also heavily harvested thus providing further examples on how whaling can produce a 'genomic footprint' as stated in the title. At least, the results of this paper should be further incorporated in the discussion, as they performed some of the same analysis. Thus, the overall findings and conclusions will be much more robust for assessing overall management and conservation of the species after past harvesting, and the paper will be much stronger.

*Line 86-88. One of the things that confused me is this statement considering the highly migratory behaviour of Fin whales (from Artic/Antarctic to tropical regions) that made me wonder if Gulf of California whales would be hunted in other places. After a quick literature search I found that this population is resident (non-migratory, e.g. assessed using satellite telemetry) which is surprising and relevant to the study. It also explains its genetic isolation (references 28-30 already cited here) and potential local adaptation. I suggest to mention it somewhere here in the introduction and again in the discussion of some of the results.

*Line 131 (this section): I was wondering if the authors explored the potential presence of structural variants based on ROHs results. With a reference genome of the same species (and thus there is no need to assume synteny and you have reference alleles for the same species to compare with) it is feasible to detect blocks of limited recombination (e.g. blocks of reference vs blocks of derived alleles) suggesting the presence of putative inversions. I am mentioning this particularly for the medium-long ROHs found in GOC whales.

*Figure 3C: Mention somewhere in the legend that the ENP population is the green one and the GOC population is the orange one. Another option is to add the code of the populations in the bottom of the respective bars in the figure.

*Lines 248-295, Figure 4. I recommend not to assume that the nonsynonymous derived alleles are deleterious (e.g. use DEL in the figure captions, or directly talking about deleterious variation in the text) and consider them just as 'potentially under selection'. Consider that these are not "the novo" mutations happening in one single individual but alleles that are found at certain frequency across all samples (in part due to the filters applied) and thus have been in the populations for some time. Additionally, the fact that both 'derived' and 'reference' alleles are found in homozygosis in certain abundance and with clear patterns across populations (Fig 4) may be explained by other population genetic processes such as genetic drift due to isolation and/or local adaptation (which will explain why you find significantly more DEL alleles in GOC in Fig 4B and Fig 4C but not in the other type of alleles, because of the combination of isolation plus local adaptation). Also consider that, under local adaptation, the selection coefficients of the alleles do not need to be the same across populations, nor even in the same direction (e.g. Fig 4c). Probably you will find these same loci by doing an FST outlier analysis looking for local adaptation. Finally, the alleles that are more probable to have a deleterious effect (LOF) are very infrequent (Fig 4B) with no differences across populations, except on the distribution of the genotypes (Fig 4A). I am not questioning the methods nor the results, just the terminology used and I also recommend to be wider in the discussion considering alternatives to the patterns found (not only deleterious variation). This is something that I generally find lacking in researchers using this approach: this is an amazing technique but, In my opinion, a wider discussion can be built on the results.

*Line 572-> Table S1 is not the comparison between the different Mysticeti genomes. It should be Table S15 (I suggest also to add the genome code (NCBI or similar) to know which version of the

genome is being used for the comparison). Also see my comment on the new Fin whale genome.

*Line 601. The monomorphic loci included only the ones that are monomorphic across all samples but different to the reference (e.g. potential species-specific fixed positions in relation to *B. acutorostrata*) or all monomorphic positions, even if they are similar to the reference (e.g. potentially covering all the genome, except low quality regions)?

*Line 608-609. The writing of the expected allele balance filtering is confusing, I suggest to replace the text in brackets by: (the following thresholds of the reference allele were used: ≥ 0.9 for homozygous reference genotypes; between ≥ 0.2 and ≤ 0.8 for heterozygous genotypes and ≤ 0.1 for homozygous alternative genotypes).

*Lines 613-617. I could not track the number of usable loci/sites after all the filtering, nor in the results or in the methods, used for each particular analysis (as different analysis used different filtered data) with some exceptions (e.g. line 633 or 639). I suggest to detail this information in a similar way for the remaining analysis (e.g. variant annotation, ADMIXTURE, runs of homozygosity...), starting with the "f50b4" dataset, the "genotype-filter-free" dataset and the initial filtered dataset (no name was given to this dataset, but it should be the output labelled as PASS in Fig S21A).

REVIEWER COMMENTS

Reviewer #1 (Remarks to the Author):

The authors have addressed all my comments including the performance of new analyses. I remain with three comments:

I find the claim in the discussion for purging to have enabled population persistence still a bit strong given that this is purely based on simulations of 0.4% of the genome (10 Mb, with lots of assumptions).

R1: We have changed the wording in the Results and Discussion of the main text (Lines 337 – 338, 421 – 422, 425 – 426, 438 – 440) in order to tone down our assertion by reducing the use of the term “purging” or using it as “potential purging”.

*“However, this increase in genetic load appears to be counteracted **by the removal of recessive strongly deleterious mutations**”* – Lines 337 – 338.

*“Population persistence in the GOC also appears to be enabled in part by **eliminating strongly deleterious mutations**”* – Lines 421 – 422.

*“However, we were unable to detect **this decrement** in our empirical dataset”* – Lines 425 – 426.

*“... and to determine the key role that gene flow and **potential purging** of deleterious variants play in the persistence of small isolated populations”* – Lines 438 – 440.

I am not sure to understand, why LOF should not be nonsynonymous mutations and hence there should be an overlap between the LOF and nonsynonymous deleterious mutations? In my eyes, a stop-gained mutation is a clear example of a nonsynonymous mutation.

R2: We thank the reviewer for the opportunity to clarify the distinction between LOF and nonsynonymous mutations. In short, there is no overlap between the loss-of-function (LOF) and putatively deleterious nonsynonymous mutations (DEL).

To give a more detailed explanation, the reviewer is right that a stop-gained mutation indeed changes the original amino acid to a stop codon, i.e., makes the mutation not synonymous. However, the fitness effect of the change to a stop codon, which causes a premature termination of a protein, compared with the change to another amino acid, is likely much more deleterious. Therefore, genome annotation standards and popular genome annotation software distinguish stop gained and amino-acid changing (strictly nonsynonymous) mutations into different categories. In our study, we utilized two commonly used software packages that make

this categorization: SIFT and SnpEff. Further, many studies in molecular evolution and population genetics typically treat nonsynonymous and LOF mutations separately, with the rationale that they have different fitness effects. For example, see Xue et al. 2015, Grossen et al. 2020, Dussex et al. 2021, Robinson et al. 2022.

In our study, we adopted the gold-standard definition in MacArthur et al. 2012, where the authors surveyed Loss-of-Function variants in human protein-coding genes, to distinguish between the two major types of not synonymous mutations: 1) nonsynonymous mutations that produce a different amino acid (which could be further classified into tolerated nonsynonymous and deleterious nonsynonymous); and 2) the mutations that cause loss of function of protein coding genes.

Below we also provide an illustration of the relationship between the four main types of mutations shown in Fig. 4 of our main text: SYN, TOL, DEL and LOF.

Regarding the formula $L = 100/2tr$:

I still don't understand, why this is not exponential. In my eyes this function should say $L = 100/(2r)^t$, because at each generation the length is halved?

R3: Thank you for following up on this issue, we might not have been clear enough. Statistically, if we describe all the ROH lengths in a genome as a random variable L , then L follows an exponential distribution: $L \sim \text{Exponential}()$. The mean of this exponential distribution of ROH lengths has been derived to be $E[L] = 100/2tr$. We cannot describe the mean of ROH lengths to be $E[L] = 100/(2r)^t$, because the recombination events are occurring randomly, but not in the middle of a chromosome each generation. This is an intuitive reasoning, for the detailed mathematical deduction on $E[L]$ see Haldane (1919). We modified the main text to clarify this (Lines 609 – 616).

“The length of ROH associated with inbreeding (L) decreases due to recombination in each generation and follows an exponential distribution^{92–94}. The mean length of ROH in the

exponential distribution is $E[L] = 100/2tr$, where $E[L]$ is the mean ROH length (in Mb), the constant 100 represents large segments belonging to the common ancestor in cM, t is the number of generations to the common ancestor and r is the assumed constant recombination rate of 1 cM/1Mb^{42,95}. Therefore, we calculated on average how many generations ago two haplotypes shared a common ancestor in each of the ROH categories as $t=100/2E[L]r$ ⁴².” – Lines 609 – 616.

And a comment only regarding the author responses to one of my comments (I am fine with them not reanalyzing using genotype likelihood approaches):

I've never read that 20x coverage (still not very high and errors are still quite likely) could be too high for analyses using genotype-likelihoods. Other than the analyses potentially running slower, I don't foresee a problem.

R4: We agree with the reviewer. We have not found any peer review publication specifically mentioning that data with higher than 20x coverage might not be properly analyzed using genotype-likelihood methods. However, to the best of our knowledge, we also have not found publications in which genotype-likelihood methods are used to analyze datasets close to 30x coverage (our data has 27x average coverage). In addition, some of the developers of genotype-likelihood methods have mentioned in public forums the possibility that this type of method might not be the best to analyze high coverage sequencing data: “I would recommend to use ANGSD if you have low coverage sequencing data. If you have high coverage data then it should be possible to call genotypes without most of the bias that exists with low coverage data. ANGSD is not meant for analysis based on called genotypes.” (<https://github.com/ANGSD/angsd/issues/131>). These are the reasons we mentioned that high coverage sequencing data might not be best analyzed by genotype-likelihood methods, but of course this conclusion might not be accurate.

Reviewer #2 (Remarks to the Author):

The authors use high coverage whole genome sequencing data on 50 individuals from two populations to unveil their demographic history considering their historical levels of whaling (non harvested population vs harvested population). This is the first time I had the opportunity to review this interesting paper, as I did not participate in the previous round of revisions. I found the paper very interesting and uses standard, albeit cutting edge, population genomic analyses to inform management and conservation of these two Fin whale populations. I have also checked the comments and answers of the prior two Referees and I mostly agree with their observations. I am also generally happy in how the authors have addressed the points raised by the prior Referees. However, in my opinion there are still some issues that needs to be solved that probably will need a major reanalysis of the data. Find below my detailed comments.

Detailed comments:

*I agree with Referee#2, who mentioned that the lack of a reference genome may be an issue, especially when aiming at high impact journal. It is true that the use a reference genome, that is not of the target species, is commonly used in intraspecific population genomic studies, although its potential biases and assumptions are rarely discussed (synteny needs to be assumed, intraspecific polymorphism may be underestimated due to the difference between the intraspecific divergence and the divergence to the reference, definition of derived vs reference alleles). I have checked the reference genomes available and I have found that a reference genome of the target species has been recently published (Wolf et al. 2022), as well as another set of whole genome sequencing of 51 individuals from North Atlantic populations (in a paper that is relatively similar to this one but in another region). I was very surprised to found this publication in your reference list (ref 25) but not the reference genome of this publication in your comparisons looking on what genome to use (e.g. Fig S20 or Table S15), despite its apparent high quality (e.g. BUSCO Cetartiodactyla C:83.4%[S:82.1%,D:1.3%] F:4.1%,M:12.5%, N50 = 2,49xE07, L50=27, 1.7xE07 genes). I would strongly advise to reanalyse your data with this new reference genome, instead of using the genome of a species that diverged between 11.5 and 17.6 MYA. You may also consider to include the data of these populations (despite being ~10 fold), as they have been also heavily harvested thus providing further examples on how whaling can produce a 'genomic footprint' as stated in the title. At least, the results of this paper should be further incorporated in the discussion, as they performed some of the same analysis. Thus, the overall findings and conclusions will be much more robust for assessing overall management and conservation of the species after past harvesting, and the paper will be much stronger.

R5: We have partially reanalyzed our data using as reference the fin whale genome the reviewer suggested (GCA_023338255.1) and compared the results with the data presented in our manuscript when the minke whale genome is the reference. To accomplish this reanalysis, we randomly chose 10 individuals to be reanalyzed (5 individuals per population; ENP and GOC). We performed all the steps in our genotyping pipeline (i.e., filtering reads, aligning to the reference genome, variant calling and filtering genotypes), as we previously did with the minke whale reference genome. The only step we did not perform was the snpEff and SIFT annotation, because this step requires a genome annotation, and the annotation for this new fin whale reference genome is not publicly available. This analysis led to relatively similar results for the mapping rate, average number of called sites and average number of heterozygous sites compared to our pervious use of the minke whale genome as the reference (Table S2).

Also, we compared the genome-wide heterozygosity and the total length of runs of homozygosity when using the fin whale or the minke whale genome as reference. We observed that there is no statistical difference in these diversity parameters when calculated using the fin whale reference genome (Fig. S1). These results indicate that using the fin whale genome as a reference will not change our results significantly.

In addition, although the fin whale genome assembly suggested by the reviewer is available, its genome annotation (which we would need to carry out the majority of our analyses of coding variation and neutral region identification) is not publicly available (https://ftp.ncbi.nlm.nih.gov/genomes/genbank/vertebrate_mammalian/Balaenoptera_physalus/

atest_assembly_versions/GCA_023338255.1_SBiKF_Bphy_ph2/). We have mentioned this in the Methods section of the main text (Lines 487 – 490).

“We used the minke whale genome as a reference because the available fin whale genome assemblies are much more fragmented and poorly annotated (GCA_008795845.1; Scaffold N50: 871,016) or they did not have a publicly available genome annotation as of November 2022 (GCA_023338255.1),” – Lines 487 – 490.

Also, according to the NCBI genome website the statistics given by the reviewer for this annotation are slightly worse than for the annotation of the minke whale genome we used.

- Fin whale genome - BUSCO Cetartiodactyla C:83.4%[S:82.1%,D:1.3%]
F:4.1%,M:12.5%, N50 = 2.49xE07, L50=27, 17,307 genes.
- Minke whale genome - BUSCO Cetartiodactyla C:97.6%[S:96.3%,D:1.4%]
F:1.1%,M:1.3%, N50 = 1.28xE07, L50=57, 26,806 genes
(https://www.ncbi.nlm.nih.gov/data-hub/genome/GCF_000493695.1/)

The abbreviations used above to describe the BUSCO scores are as follow: (C)omplete and (S)ingle; (C)omplete and (D)uplicated; (F)ragmented and (M)issing BUSCO genes. N50: scaffold N50; L50: scaffold L50. Number of genes compiled from NCBI report for the minke whale genome and from the Wolf et al. 2022 publication for the fin whale genome.

Furthermore, the potential biases that the reviewer mentions do not seem to affect our results either. Synteny across large regions of the genome is not needed for the analysis of coding regions and might not be expected to play an important role when the mapping rate is high (99.09% reads mapped using minke whale) and the focus is on analyzing SNPs, which is the case in our study. However, ROH analysis might be affected by a lack of synteny, but given that there are not differences in total ROH length between reference genomes, it seems synteny is not a problem in our study. Even if intraspecific polymorphism could be slightly underestimated, it does not seem to affect the accurate estimation of genome-wide heterozygosity and ROH as shown in our analyses described above. Finally, we have acknowledged in the Methods section (Lines 542 – 546) that: *“Because the minke whale has evolved since the common ancestor with these two populations of fin whales, the ancestral alleles identified may not represent the true ancestral state. However, this error is not expected to bias the relative comparison of variants between the ENP and GOC fin whales since they are equally diverged from the minke whale.”*. Similar to our findings, another study that evaluated the reference genome choice (ferret vs sea otter, 10 My divergence) found no significant differences for heterozygosity, ROH and demographic inference in sea otters (Beichman et al. 2022).

The results of this new analysis with the fin whale genome and the above discussion show that the potential biases caused by using the minke whale reference genome are minimal. We incorporated the comparison analyses described above and the results in the Methods and Results sections of the main text (Lines 111 – 116, 493 – 497, 530 – 534). Additionally, to give more details about these analyses and their results, we added a new section in the Supplemental Methods called “Genotyping, Heterozygosity and ROH using a fin whale reference genome” (Supplemental text lines 5 – 25) and a section “Comparing genotyping and

diversity estimates using the minke and fin whale reference genomes” in the Supplemental Results (Supplemental text lines 78 – 90), including Fig. S1 and Table S2 as previously mentioned.

“We also genotyped a subset of ten individuals using a recently available fin whale genome assembly (GCA_023338255.1). Both reference genomes provide similar genotyping statistics and genomic diversity results (Table S2; Fig. S1; Supplemental Methods and Results), suggesting that using the minke whale genome as a reference does not introduce significant biases in our analyses (see discussion and significance tests in Supplemental Results).” – Lines 111 –116.

“The average mapping rate of fin whale reads to the minke whale genome is $99.09 \pm 0.21\%$ (Table S1), which is similar to the 99.49% mapping rate to the most recent fin whale reference genome (GCA_023338255.1; Table S2), obtained from a subset of samples ($n=10$; see Supplemental Methods), suggesting that the divergence time with minke whales did not strongly impact read alignment.” – Lines 493 – 497.

“We also performed the same genotyping pipeline using the most recent fin whale genome as reference (GCA_023338255.1) in a subset of 10 individuals (10-fin-ref dataset) to determine if there were significant differences in genomic diversity estimates caused by the reference genome used (minke whale vs fin whale; see Supplemental Methods and Results)” – Lines 530 – 534.

“Genotyping, heterozygosity and ROH using a fin whale reference genome

In population genomic studies, using the genome from a species that is not the focal species as reference could potentially be problematic if both species diverged long ago because they have accumulated genetic variation independently and might not be useful to detect variation in the other species. To determine if using the minke whale genome as reference in our fin whale dataset would cause a substantial difference in genotyping statistics and in genomic diversity estimations (i.e. genome-wide diversity and runs of homozygosity (ROH)), we used a fin whale genome that has been made recently available (GCA_023338255.1). However, its annotation is not publicly available, preventing us from using it as the primary reference genome in this study. We randomly selected a subset of 10 fin whale samples [5 per population (ENP: ENPAK24, ENPAK30, ENPCA04, ENPCA08 and ENPWA15; GOC: GOC006, GOC050, GOC071, GOC086, GOC100)] to perform these analyses (we named this dataset 10-fin-ref). First, we performed all the steps of our genotyping pipeline using the fin whale genome GCA_023338255.1, except the snpEff and SIFT annotation step that requires a genome annotation. Second, with the genotyping data obtained using the fin whale reference genome, we calculated the genome-wide heterozygosity for the 10 individuals as previously mentioned in the main manuscript, and compared them with the results obtained with the data using the minke whale reference genome. We performed a Wilcoxon test to determine if the differences observed were statistically significant due to the reference genome choice. Finally, we calculated the total ROH length with the bcftools software and compared them with the results

obtained using the minke whale genome as reference and performed a Wilcoxon test.” – Supplemental text lines 5 – 25.

“Comparing genotyping and diversity estimates using the minke and fin whale reference genomes

We found that there were not great differences in the statistics from the genotyping pipeline when using the minke whale or fin whale genome as the reference (Table S2). For example, the average mapping rate to both genomes were higher than 99% in both cases (minke whale = 99.09%, fin whale = 99.49%) with the fin whale genome having a slightly higher mapping rate, as expected. The average number of heterozygous sites identified were relatively similar (minke whale = 1,290,413, fin whale = 1,457,881). No significant differences between the two reference genomes were found in genome-wide heterozygosity (Figs. S1A, S1B) or total ROH length (Figs. S1C, S1D). These results indicate that regardless of the reference genome used, the genotyping pipeline results and the genomic diversity estimates are similar, suggesting that using the minke whale genome as a reference does not significantly affect the results of our analyses.” – Supplemental text lines 78 – 90.

Also, as suggested by the reviewer, we included this new fin whale genome’s statistics in a supplementary table (Table S16), and further discussed the results of the Wolf et al. 2022 publication in the Results and Discussion sections of the main text (Lines 266 – 269, 365 – 369, 382 – 384, 386 – 388, 400 – 402).

“For all four mutation types, heterozygosity is significantly depleted and homozygosity is significantly elevated in the GOC population (MWU tests $P = 2.9E-12$ in all comparisons; Table S15). This pattern has not been reported in other fin whale populations or great whale species²⁵ and is consistent with reduced genome-wide heterozygosity and small population size” – Lines 266 – 269.

“Recently, low-coverage sequencing of North Atlantic fin whales may have recovered a signal of whaling, although the results did not completely rule out the alternative scenario of a more gradual decline over the last 600 years rather than an abrupt whaling bottleneck²⁵, two scenarios which are challenging to disentangle, particularly with added uncertainties associated with low-coverage data.” – Lines 365 – 369.

“Our study is one of the first to examine the natural experiment of whale populations that have experienced both natural and anthropogenic population bottlenecks, providing unique contrasts not available in single-population studies²⁵.” – Lines 382 – 384.

“They do not exhibit a substantial decrease in genome-wide heterozygosity nor an increase in inbreeding or genetic load (Figs. 2, 4 and 5A), similar to what was found in a North Atlantic population²⁵.” – Lines 386 – 388.

“These simulations allowed us to explore genomic consequences under various conservation scenarios (Fig. 5), an important perspective not yet adopted in other great whale genomic studies^{25,38,59}” – Lines 400 – 402.

Finally, reanalyzing previously published data from the North Atlantic is not within the scope of our paper. Especially when the data has significantly less sequencing coverage compared to ours, which is precisely one of the characteristics of our data we had argued, in previous revision rounds and in our manuscript, allowed us to confidently detect the timing and severity of the whaling bottleneck and perform an accurate estimate of deleterious variation.

*Line 86-88. One of the things that confused me is this statement considering the highly migratory behaviour of Fin whales (from Artic/Antarctic to tropical regions) that made me wonder if Gulf of California whales would be hunted in other places. After a quick literature search I found that this population is resident (non-migratory, e.g. assessed using satellite telemetry) which is surprising and relevant to the study. It also explains its genetic isolation (references 28-30 already cited here) and potential local adaptation. I suggest to mention it somewhere here in the introduction and again in the discussion of some of the results.

R6: We have added the text about the resident nature of the Gulf of California population in the Introduction section (Lines 83 – 85) and mention it in the Discussion section of the main text (Lines 405 – 409).

“However, fin whales in the Gulf of California, Mexico, belong to a resident population that was not targeted by whalers^{27,28}. Nevertheless, this population has been small with limited gene flow from and to the Pacific for thousands of years²⁸⁻³¹.” – Lines 83 – 85.

“Regarding the Gulf of California fin whale population, our results show that immigration from ghost populations is negligible (see Supplemental Discussion) and as few as 0.39 migrants per generation have been sufficient to maintain genetic diversity and fitness in this population over ~16,000 years of isolation (Fig. 5B), which is consistent with other genetic and ecological studies describing the isolation of this population^{28,30,34}” – Lines 405 – 409.

*Line 131 (this section): I was wondering if the authors explored the potential presence of structural variants based on ROHs results. With a reference genome of the same species (and thus there is no need to assume synteny and you have reference alleles for the same species to compare with) it is feasible to detect blocks of limited recombination (e.g. blocks of reference vs blocks of derived alleles) suggesting the presence of putative inversions. I am mentioning this particularly for the medium-long ROHs found in GOC whales.

R7: Although the study of structural variation is undoubtedly very interesting and could be informative on potential deleterious or adaptive variation, it is not within the scope of this study. We thank the reviewer’s suggestion on this exciting venue of analyses, which we are considering exploring in a follow up study. However, given the results of our additional ROH analyses using the fin whale reference genome, we do not expect that structural variants could

impact ROH calling and proper tests to detect inversions would be needed. In the 10 individuals in which ROH were called using both the fin whale and the minke whale genome as reference, we did not observe significant differences in the total ROH length identified within each individual (Figs. S1C, S1D). See **R5** above for more detail.

*Figure 3C: Mention somewhere in the legend that the ENP population is the green one and the GOC population is the orange one. Another option is to add the code of the populations in the bottom of the respective bars in the figure.

R8: Thank you for pointing this out. We have added the explanation of the meaning of the colors in the figure caption (Lines 1194 – 1195, 1205 – 1207, 1232 – 1233, 1238 – 1241, 1243 – 1244).

*“(A) The historical demography of the Eastern North Pacific (ENP; **green**) population”* – Lines 1194 – 1195.

*“After the divergence, the ENP population (**green**) remained at an effective population size of ~17,000, whereas the Gulf of California (GOC; **orange**) population has remained small at an effective size of $N_e = 114$.”* – Lines 1205 – 1207.

*“Results for simulations under single-population 3-epoch model for the ENP population (**green**)”*
Lines 1232 – 1233.

*“Results for simulations under our chosen two-population model. Each quantity is shown for the ENP (**green**) and GOC (**orange**; GOC w/mig) populations at the end of the simulation. We also simulated under a no migration demographic scenario for the GOC population (**orange**; GOC w/o mig).”* Lines 1238 – 1241.

*“In the demographic representations, the sampled population, ENP or GOC, are shown in **green** or **orange**, respectively,”* Lines 1243 – 1244.

*Lines 248-295, Figure 4. I recommend not to assume that the nonsynonymous derived alleles are deleterious (e.g. use DEL in the figure captions, or directly talking about deleterious variation in the text) and consider them just as ‘potentially under selection’.

R9: Thanks for the detailed comments. To facilitate the readability and clearness of our responses to the points mentioned here, we reply to each of them in separate paragraphs.

We agree that no assumptions should be made about nonsynonymous derived alleles being deleterious. To avoid this assumption, we used the categorization implemented by the program SIFT4G to sort the putatively deleterious nonsynonymous variants from the putatively tolerated nonsynonymous variants (see the illustration above in **R2**) given various degrees of phylogenetic constraint (i.e., nonsynonymous variants occurring at conserved regions are more likely to be deleterious) (Kumar et al. 2009; Vaser et al. 2016). The accuracy of the SIFT

predictions has been demonstrated from human disease datasets to bacteriophage datasets, where SIFT correctly predicts more than 70% of nonsynonymous variants assigned as “deleterious” are associated with the disease and affect gene function (Kumar et al. 2009; Vaser et al. 2016). In addition, it has been well established that most non-synonymous variants have deleterious effects (Eyre-Walker and Keightley 2007, Boyko et al. 2008). Moreover, we are careful in interpreting the nonsynonymous variants assigned as “deleterious” by SIFT and referred to them as “putatively deleterious” variants throughout our main text (Lines 252, 256 – 258, 259 – 261, 270, 273 – 274, 278 – 279, 299 – 301, 426 – 427, 806, 807 – 808, 811 – 813, 825 – 826, 1214, 1216 – 1217, 1218 – 1220). For example:

*“**Putatively** deleterious variation and genetic load”* – Line 252.

*“The derived alleles were classified into four mutation types: synonymous, tolerated nonsynonymous (SIFT score ≥ 0.05), **putatively** deleterious nonsynonymous (SIFT score < 0.05), and loss-of-function”* – Lines 256 – 258.

*“The synonymous and tolerated nonsynonymous mutations serve as a proxy for neutral variants whereas the **putatively** deleterious nonsynonymous and LOF variants are proxies for **putatively** deleterious variants⁴⁸”* – Lines 259 – 261.

*“as well as an overall accumulation of **putatively** deleterious nonsynonymous compared to the ENP population”* – Lines 299 – 301.

*“Two lines of evidence were used to quantify relative levels of **putatively** deleterious variation”* – Lines 807 – 808.

*“The nonsynonymous mutations were classified as **putatively** tolerated (SIFT score ≥ 0.05) or deleterious (SIFT score < 0.05) based on phylogenetic constraints using SIFT4G⁸³”* – Lines 811 – 813.

We understand that in this comment the reviewer referred the nonsynonymous variants to “potentially under selection” because there are considerations of local adaptation processes. We addressed this point in **R11** below.

Consider that these are not “the novo” mutations happening in one single individual but alleles that are found at certain frequency across all samples (in part due to the filters applied) and thus have been in the populations for some time.

R10: We are not sure if we completely understood the reviewer’s comment here. We fully agree that the variants found in our dataset are not de novo mutations. We referred to mutations broadly in the same way as the reviewer described them in their comment. Indeed, in general, de novo nonsynonymous mutations would be more deleterious compared to standing nonsynonymous alleles that have been in the population for some time (the mutations in our dataset), because purifying selection more efficiently removes highly deleterious amino acid

mutations from the population while neutral or slightly-deleterious/nearly-neutral variants can persist. However, these standing variants are more likely to be neutral or slightly-deleterious/nearly-neutral rather than being adaptive, because positive selection/local adaptation would quickly fix the adaptive/beneficial mutation in the population, removing most of them from the standing variations we are analyzing here. In fact, of all 116,908 variants analyzed in coding regions (reported in Table S15), we only found one variant in which the derived allele is fixed in the GOC population when the reference allele is fixed in the ENP population (all genotypes were 1/1 in GOC and 0/0 in ENP at that site).

Additionally, the fact that both ‘derived’ and ‘reference’ alleles are found in homozygosis in certain abundance and with clear patterns across populations (Fig 4) may be explained by other population genetic processes such as genetic drift due to isolation and/or local adaptation (which will explain why you find significantly more DEL alleles in GOC in Fig 4B and Fig 4C but not in the other type of alleles, because of the combination of isolation plus local adaptation). Also consider that, under local adaptation, the selection coefficients of the alleles do not need to be the same across populations, nor even in the same direction (e.g. Fig 4c). Probably you will find these same loci by doing an F_{ST} outlier analysis looking for local adaptation.

R11: We agree that genetic drift, purifying (negative) selection and/or local adaptation are important population genetic processes. We state this in our manuscript (Lines 54 – 56):

“Both anthropogenic and naturally occurring population declines reduce genetic diversity, increase inbreeding and genetic load due to the stronger action of genetic drift which diminish the long-term survival and adaptive potential of populations^{7,8}.” – Lines 54 – 56.

Also, we explain “why we find significantly more DEL alleles in GOC” by demonstrating the potential relative forces of genetic drift and purifying selection in the small GOC population (Lines 282 – 284):

“the small population size of the GOC population likely increased the strength of genetic drift and decreased the efficacy of selection compared to the larger ENP population,” – Lines 282 – 284.

Moreover, although positive selection could be acting on the GOC population, the general pattern of our results is highly consistent with a stronger action of genetic drift and a diminished role of purifying selection leading to an accumulation of slightly and moderately deleterious variants in the GOC population, but not local adaptation. Briefly, local adaptation (positive selection) would not cause an increase in homozygosity at a genome-wide scale in all mutation categories (i.e., synonymous as well as nonsynonymous (tolerated and deleterious) and LOF) as observed in Figs. 4A and 4C, nor is it a very reasonable explanation for an accumulation of the total number of nonsynonymous mutations, given that the GOC has had a small N_e for hundreds of generations. Furthermore, as mentioned before, most amino-acid changing mutations are deleterious (Eyre-Walker and Keightley 2007, Boyko et al. 2008), therefore, adaptive processes are considerably less prevalent compared to genetic drift and purifying

selection, which are always present (Johri et al. 2020; Zhang et al. 2020; Johri et al. 2022). To emphasize this point, we added more explanation in this section of the results in the main text, with more explicit mention of local adaptation and genetic drift (Lines 261 – 263, 282 – 284):

“Although amino-acid changing variants could serve as candidates for local adaptation, most of them are deleterious^{49,50}. ” — Lines 261 – 263.

“Assuming that these nonsynonymous alleles are slightly deleterious, the small population size of the GOC population likely increased the strength of genetic drift and decreased the efficacy of selection compared to the larger ENP population,” – Lines 282 – 284.

It is to be expected that a subset of mutations might be under selection due to local adaptation, but it would require a proper test for selection to differentiate sites under selection versus those evolving via drift. An F_{st} outlier test might not be a good test for local adaptation because it does not distinguish between drift and selection. Detecting local adaptation is not within the scope of our study but it certainly is an interesting follow up study that we are considering to pursue thanks to the reviewer’s comments.

Finally, the alleles that are more probable to have a deleterious effect (LOF) are very infrequent (Fig 4B) with no differences across populations, except on the distribution of the genotypes (Fig 4A).

R12: Yes, the LOF alleles are expected to be infrequent and the numbers of LOF alleles or heterozygotes are comparable to similar studies. We identified that on average, the Eastern North Pacific fin whales contain 159 LOF heterozygotes whereas the Gulf of California individuals contain 96 (Table S15), which is on the same scale of 50 to 250 LOF heterozygotes reported in Fig. 2D in Robinson et al. (2022) for 12 cetacean species with different demographic histories. With regards to the finding that there are no differences in LOF alleles, we explain this observation in the Results section in our revised manuscript (Lines 282 – 287):

“Assuming that these nonsynonymous alleles are slightly deleterious, the small population size of the GOC population likely increased the strength of genetic drift and decreased the efficacy of selection compared to the larger ENP population, allowing the persistence of deleterious variants in the Gulf. By contrast, the similar number of LOF alleles indicates that, in spite of the GOC population’s small size, purifying selection has remained effective at eliminating the most deleterious mutations.” – Lines 282 – 287

I am not questioning the methods nor the results, just the terminology used and I also recommend to be wider in the discussion considering alternatives to the patterns found (not only deleterious variation). This is something that I generally find lacking in researchers using this approach: this is an amazing technique but, In my opinion, a wider discussion can be built on the results.

R13: Thank you for the suggestions. Hopefully the edits described above clarify our results and expand the discussion more widely to address the insightful points you have raised above.

Additionally, we added some sentences in the discussion section of the main text about local adaptation as suggested by the reviewer (Lines 429 – 435):

“Although it could be argued that some genomic patterns of deleterious variation might reflect local adaptation in the GOC population, this explanation seems unlikely. For example, only drift would cause increased homozygosity in all mutation categories as observed, specifically, increased homozygosity in synonymous variants is not expected under a scenario of local adaptation (Fig. 4A, 4C). Moreover, local adaptive events occur more rarely than genetic drift and purifying selection that are constantly ongoing in natural populations⁷⁰.” – Lines 429 – 435.

*Line 572-> Table S1 is not the comparison between the different Mysticeti genomes. It should be Table S15 (I suggest also to add the genome code (NCBI or similar) to know which version of the genome is being used for the comparison). Also see my comment on the new Fin whale genome.

R14: Thanks for pointing this out. However, here we intended to refer to Table S1 but not Table S15. At the bottom of Table S1, we show the data for the four mysticete resequencing fastq SRA mentioned. In this paragraph, we have now clarified that we have downloaded whole-genome resequencing fastq data from NCBI SRA for four representative Mysticeti species to compare genome wide heterozygosity (Fig. 2A). We were not presenting methods for genome assembly comparisons, which is detailed in the Supplemental Methods and now shown in Table S16. We have modified the main text to clarify this difference (Lines 473 – 475):

“To compare the fin whales’ genomic characteristics within Mysticeti, previously generated whole-genome resequencing fastq data from four representative Mysticeti species were downloaded from the NCBI Sequence Read Archive” – Lines 473 – 475.

*Line 601. The monomorphic loci included only the ones that are monomorphic across all samples but different to the reference (e.g. potential species-specific fixed positions in relation to *B. acutorostrata*) or all monomorphic positions, even if they are similar to the reference (e.g. potentially covering all the genome, except low quality regions)?

R15: Sorry for not clarifying this distinction in the main text. We referred to the latter, i.e., monomorphic loci include all monomorphic positions, even if they are similar to the reference (e.g. potentially covering all the genome, except low quality regions). We have modified the main text to clarify this distinction (Lines 509 – 512):

“Instead, we performed a stringent set of quality and depth filters for the genotype calls, keeping only high-quality biallelic SNPs and monomorphic sites with the latter including all homozygous reference or all homozygous alternate genotypes (Fig. S22).” – Lines 509 – 512.

*Line 608-609. The writing of the expected allele balance filtering is confusing, I suggest to replace the text in brackets by: (the following thresholds of the reference allele were used: ≥ 0.9 for homozygous reference genotypes; between ≥ 0.2 and ≤ 0.8 for heterozygous genotypes and ≤ 0.1 for homozygous alternative genotypes).

R16: We have modified the text according to your suggestion (Lines 519 – 522):

“(the following thresholds were used for the allele balance, defined as the read depth for the reference allele divided by the total read depth: ≥ 0.9 for homozygous reference genotypes; between ≥ 0.2 and ≤ 0.8 for heterozygous genotypes; and ≤ 0.1 for homozygous alternative genotypes)”. – Lines 519 – 522.

*Lines 613-617. I could not track the number of usable loci/sites after all the filtering, nor in the results or in the methods, used for each particular analysis (as different analysis used different filtered data) with some exceptions (e.g. line 633 or 639). I suggest to detail this information in a similar way for the remaining analysis (e.g. variant annotation, ADMIXTURE, runs of homozygosity...), starting with the “f50b4” dataset, the “genotype-filter-free” dataset and the initial filtered dataset (no name was given to this dataset, but it should the output labelled as PASS in Fig S21A).

R17: Thanks for the suggestion. We have included the following supplementary table (Table S17) to aid readers navigating the number of usable sites and the dataset used in the study.

Dataset	Samples	Filtration method	Analyses in which the dataset was used	Number of sites that passed all filters
all50	All 50 fin whales	Standard	All the analyses except those mentioned for the other datasets below.	890,858,824
f50b4	All 50 fin whales + four Mysticeti species	Standard	Genome wide heterozygosity for four Mysticeti species (Fig.2 and Fig. S8). Neighbor-joining tree (Fig. S5).	880,177,286
genotype-filter-free	All 50 fin whales	No genotype filters	Demographic inference using the SFS without genotype filtering (Table S10).	934,524,879
10-fin-ref	Subset of 10 samples (five per population; see	Standard	Complementary analyses of genome-wide heterozygosity and total	1,084,268,877

Cited References

- Beichman, Annabel C., Pooneh Kalhori, Christopher C. Kyriazis, Amber A. DeVries, Sergio Nigenda-Morales, Gisela Heckel, Yolanda Schramm, Andrés Moreno-Estrada, Douglas J. Kennett, Mark Hylkema, James Bodkin, Klaus-Peter Koepfli, Kirk E. Lohmueller, and Robert K. Wayne. 2022. “Genomic Analyses Reveal Range-wide Devastation of Sea Otter Populations.” *Molecular Ecology*, In press. <https://doi.org/10.1111/mec.16334>.
- Boyko AR, Williamson SH, Indap AR, Degenhardt JD, Hernandez RD, Lohmueller KE, et al. 2008. Assessing the Evolutionary Impact of Amino Acid Mutations in the Human Genome. *PLoS Genetics*, 4(5): e1000083. <https://doi.org/10.1371/journal.pgen.1000083>
- Dussex N, Van Der Valk T, Morales HE, Wheat CW, Díez-del-Molino D, Von Seth J, Foster Y, Kutschera VE, Guschanski K, Rhie A, Phillippy AM. et al. 2021. Population genomics of the critically endangered kākāpō. *Cell Genomics*, 1(1): 100002.
- Eyre-Walker, A., Keightley, P. 2007. The distribution of fitness effects of new mutations. *Nature Review Genetics*, 8: 610–618. <https://doi.org/10.1038/nrg2146>
- Grossen, C., Guillaume, F., Keller, L. F. & Croll, D. 2020. Purging of highly deleterious mutations through severe bottlenecks in Alpine ibex. *Nature Communications*, 11: 1–12.
- Haldane, J.B.S. 1919. The combination of linkage values, and the calculation of distances between the loci of linked factors. *Journal of Genetics*, 8: 299-309.
- Johri P, Charlesworth B, Jensen JD. 2020 Toward an Evolutionarily Appropriate Null Model: Jointly Inferring Demography and Purifying Selection. *Genetics*, 215(1): 173–192. <https://doi.org/10.1534/genetics.119.303002>
- Johri P, Aquadro CF, Beaumont M, Charlesworth B, Excoffier L, Eyre-Walker A, Keightley PD, Lynch M, McVean G, Payseur BA, Pfeifer SP, Stephan W, Jensen JD. 2022. Recommendations for improving statistical inference in population genomics. *PLoS Biology*, 20(5): e3001669. <https://doi.org/10.1371/journal.pbio.3001669>
- Kumar, P., Henikoff, S. & Ng, P. 2009. Predicting the effects of coding non-synonymous variants on protein function using the SIFT algorithm. *Nature Protocols*, 4: 1073–1081. <https://doi.org/10.1038/nprot.2009.86>
- MacArthur DG, Balasubramanian S, Frankish A, Huang N, Morris J, Walter K, Jostins L, Habegger L, Pickrell JK, Montgomery SB, Albers CA, et al. 2012. A systematic survey of loss-of-function variants in human protein-coding genes. *Science*, 335(6070): 823-828. DOI: [10.1126/science.1215040](https://doi.org/10.1126/science.1215040)
- Robinson JA, Kyriazis CC, Nigenda-Morales SF, Beichman AC, Rojas-Bracho L, Robertson KM, Fontaine MC, Wayne RK, Lohmueller KE, Taylor BL, Morin PA. 2022. The critically endangered vaquita is not doomed to extinction by inbreeding depression. *Science*, 376(6593): 635–639. DOI: [10.1126/science.abm1742](https://doi.org/10.1126/science.abm1742)

- Vaser, R., Adusumalli, S., Leng, S. *et al.* 2016. SIFT missense predictions for genomes. *Nature Protocols*, **11**: 1–9 . <https://doi.org/10.1038/nprot.2015.123>.
- Xue, Y. *et al.* 2015. Mountain gorilla genomes reveal the impact of long-term population decline and inbreeding. *Science* 348, 242–245.
- Zhang X, Kim B, Lohmueller KE, Huerta-Sánchez E. 2020. The Impact of Recessive Deleterious Variation on Signals of Adaptive Introgression in Human Populations. *Genetics*, 215(3): 799–812. <https://doi.org/10.1534/genetics.120.303081>

REVIEWERS' COMMENTS

Reviewer #1 (Remarks to the Author):

Comments whaling paper

The authors have addressed all my comments.

However, regarding their reanalysis addressing the comments by reviewer#2 I was a little disappointed by how little use they made of their additional work.

Rather than claiming that the observed differences were not statistically significant, they could have discussed the potential implications for their results. For instance they could have mentioned that the slightly higher diversity with the fin whale genome may be due to more confident snp calls in the less conserved regions (as expected). Hence there is maybe a slight underestimation of diversity with their overall analysis.

Also, do I understand correctly that the authors did not contact the authors of the fin whale reference genome to ask for availability of the annotation? That is a little surprising.

And there seems to be a hiccup at line 678.

Reviewer #2 (Remarks to the Author):

I am satisfied with how the authors have answered all my comments.

Response to Referees

Reviewer #1's comments:

The authors have addressed all my comments.

However, regarding their reanalysis addressing the comments by reviewer#2 I was a little disappointed by how little use they made of their additional work.

Rather than claiming that the observed differences were not statistically significant, they could have discussed the potential implications for their results. For instance they could have mentioned that the slightly higher diversity with the fin whale genome may be due to more confident snp calls in the less conserved regions (as expected). Hence there is maybe a slight underestimation of diversity with their overall analysis.

R1: We are glad we were able to address reviewer 1's comments but we disagree that we made little use of our additional work. The major implication of our additional work is that the difference in reference genome choices does not impact our results, which was the main concern of reviewer 2. To show this, we had to perform a proper statistical test that turned out to be insignificant. Therefore, mentioning that there are no significant differences in using either genome was the main result to ease reviewer 2's concerns.

We appreciate the reviewers' suggestion to further discuss the implications of the differences resulting from reference genome choices. However, we want to mention that, although more heterozygous sites were found when we mapped to the fin whale genome (Table S2), which could be interpreted as a higher diversity, we actually observed a slightly higher genome-wide diversity when using the minke whale genome but not the fin whale genome as the reference (Fig. S1A-B). This is because when mapping to the fin whale genome more sites were called, as expected (Average number of called sites; Table S2). Therefore, we do not have a slight underestimation of diversity within our overall analysis. If anything, we might have a slight overestimation of diversity when using the minke whale genome. Moreover, this slight overestimation is extremely minimal, with an average heterozygosity rate of 0.00140 (when mapping to fin whale) vs 0.00142 (when mapping to minke whale) for the 10 individuals we reanalyzed, an increase of less than 1.5%. Therefore, the bias in heterozygosity estimates is negligible. Following the reviewer's suggestion, we have incorporated some of these results in the main text (Lines 116 – 118). Additionally, we included sections in the Supplementary Results and Discussion to explicitly describe and discuss the results of the additional work we did comparing the two reference genomes. (Supplemental text lines 79 – 101, 196 – 227).

"We observed only a 1.5% overestimation of diversity when using the minke whale genome as reference, which could be due to a less accurate mapping (See Supplemental Discussion)."
Lines 116 – 118.

"Comparing genotyping and diversity estimates using the minke and fin whale reference genomes"

There were no great differences in genotyping or genetic diversity statistics when we used the minke whale or fin whale genome as the reference (Table S2). For example, the average mapping rate to both genomes were higher than 99% in both cases (minke whale = 99.09%, fin whale = 99.49%) with the fin whale genome having a slightly higher mapping rate, as expected. Although the average number of heterozygous sites identified was slightly higher when aligned to the fin whale genome (Table S2; minke whale = 1,290,413, fin whale = 1,457,881), which could be interpreted as obtaining higher genetic diversity, we observed a

slightly higher genome-wide diversity when the minke whale genome but not the fin whale genome is used as reference (Fig. S1A-B). This is because more sites were called when the fin whale genome is the reference, as expected (Average number of called sites; Table S2), resulting in a lower genome-wide number of heterozygous sites per called sites. Therefore, we do not observe a slight underestimation of diversity using the minke whale genome reference, if anything, we have a slight overestimation of diversity. This overestimation is minimal when we calculate it for the 10 individuals we reanalyzed, with an average heterozygosity rate of 0.00140 (when mapping to fin whale) vs 0.00142 (when mapping to the minke whale), an increase of less than 1.5%. Therefore, the bias in heterozygosity estimates is negligible to our main results. Additionally, no significant differences between the two reference genomes were found in genome-wide heterozygosity (Figs. S1A, S1B) or total ROH length (Figs. S1C, S1D). All these results indicate that regardless of the reference genome used, the genotyping pipeline results and the genomic diversity estimates are similar, suggesting that using the minke whale genome as a reference does not significantly affect the results of our analyses.” Supplemental text lines: 79 – 101.

“Potential biases when using a reference genome from a closely related species for variant calling.

The potential biases of using a reference genome from a closely related species instead of the focal species include a lack of synteny, the intraspecific polymorphism might be underestimated due to the difference between the divergence at the population level and the divergence to the reference taxa and the definition of derived vs reference alleles may be problematic. However, these potential biases do not seem to affect our results when using the minke whale genome as reference. Synteny across large regions of the genome is not needed for the analysis of coding regions and might not be expected to play an important role when the mapping rate is high (99.09% reads mapped using minke whale) and the focus is on analyzing SNPs, which is the case in our study. Although our ROH analysis might be affected by a lack of synteny, given that we did not observe significant differences in total ROH length between reference genomes (Fig. S1C, D; minke whale vs fin whale), it seems synteny is not a problem in our study. Even if intraspecific polymorphism could be slightly underestimated when we use the minke whale genome as a reference, we do not observe such underestimation, if anything we observe a slight overestimation due to less accurate mapping, but this does not seem to affect the estimation of genome-wide heterozygosity and ROH as shown in our analyses comparing these diversity estimates between reference genomes (Fig. S1A-D). Additionally, because the divergence time between the minke whale and the two fin whale populations is approximately the same, using its genome to identify the ancestral and reference alleles is not expected to introduce a significant bias.

The BUSCO statistics for the fin whale assembly that has been published⁹ are worse than the minke whale assembly that we used. The BUSCO statistics for the fin whale genome are as follow: BUSCO Cetartiodactyla C:83.4%[S:82.1%, D:1.3%], F:4.1%, M:12.5%, N50 = 2.49xE07, L50 = 27, 17,307 genes⁹. Whereas the BUSCO analysis reported in the NCBI website for the minke whale genome we used are: BUSCO Cetartiodactyla C:97.6%[S:96.3%, D:1.4%], F:1.1%,M:1.3%, N50 = 1.28xE07, L50 = 57, 26,806 genes (https://www.ncbi.nlm.nih.gov/data-hub/genome/GCF_000493695.1/). Therefore, even if such fin whale genome annotation was available, using an assembly of less quality could negatively impact our results, especially the accuracy of the deleterious variation analyses, and given the results of our reanalysis comparing genotyping and diversity estimates it seems that using the fin whale genome assembly and annotation will not make a significant difference for our main results and conclusions.” Supplemental text lines 196 – 227.

Also, do I understand correctly that the authors did not contact the authors of the fin whale reference genome to ask for availability of the annotation? That is a little surprising.

R2. Yes, we acknowledge that we did not reach out to the authors of the fin whale reference genome (Wolf et al. 2022) for the availability of the annotation.

We did not reach out to Wolf et al. during this round of revision because of several reasons:

1) Given the genome annotation statistics reported in Wolf et al. 2022, the annotation quality is worse than the minke whale genome we used. Therefore, using an annotation of lower quality does not seem appropriate, because it could negatively impact the quality of our results, especially for the deleterious variation analysis.

- Fin whale genome - BUSCO Cetartiodactyla C:83.4%[S:82.1%,D:1.3%] F:4.1%,M:12.5%, N50 = 2.49xE07, L50=27, 17,307 genes.

- Minke whale genome - BUSCO Cetartiodactyla C:97.6%[S:96.3%,D:1.4%] F:1.1%,M:1.3%, N50 = 1.28xE07, L50=57, 26,806 genes (https://www.ncbi.nlm.nih.gov/data-hub/genome/GCF_000493695.1/). We included this comparative data in the Supplemental Discussion (Supplemental text lines 217 – 223).

2) Given our reanalysis comparing genotyping and diversity estimates using the minke and fin whale reference genomes (detailed in Supplemental Methods and Results) which did not find significant differences due to reference genome choices, we are confident that the reference genome choice is unlikely to considerably impact our main results.

3) In addition, previous studies (Beichman et al. 2022) compared the differences in using ferret vs sea otter reference genome annotations (10 My divergence) for demographic modeling and deleterious variation profiling and found no significant differences either. This provides independent proof that using the reference genome and annotation of a relatively close species does not seem to greatly affect results of demographic and deleterious variation analyses.

4) Genome annotations are often provided together with their genome assemblies by the time of article publication. Since Wolf et al. 2022 was published relatively recently on May 05, 2022, and despite our best effort in querying the data availability statement and the NCBI genome portal, we could not find the annotation file. It was a little surprising for us too. We assume the authors have valid reasons for not publishing their annotation. Regardless of their reasons, their annotation is of less quality than the one we used and in light of our additional work it seems that waiting to get their annotation will not make a significant difference for our main results and conclusions.

Additionally, we mention more explicitly that our results might not be affected by the use of a different reference genome in the Methods of the main text (Lines 558 – 562) and supplemental material (Supplemental text lines 217 – 227).

“Although a recent fin whale genome assembly (GCA_023338255.1) has been annotated²⁵, this annotation is not publicly available at the present time, preventing us to use it to identify putatively neutral regions for our demographic and deleterious variation analyses. In addition, if the annotation of this fin whale genome assembly would be available it is unlikely it will significantly affect our main results and conclusions (See Supplemental Discussion).” Lines 558 – 562.

“The BUSCO statistics for the fin whale assembly that has been published⁹ are worse than the minke whale assembly that we used. The BUSCO statistics for the fin whale genome are as follow: BUSCO Cetartiodactyla C:83.4%[S:82.1%, D:1.3%], F:4.1%, M:12.5%, N50 = 2.49xE07,

L50 = 27, 17,307 genes⁹. Whereas the BUSCO analysis reported in the NCBI website for the minke whale genome we used are: BUSCO Cetartiodactyla C:97.6%[S:96.3%, D:1.4%], F:1.1%,M:1.3%, N50 = 1.28xE07, L50 = 57, 26,806 genes (https://www.ncbi.nlm.nih.gov/data-hub/genome/GCF_000493695.1/). Therefore, even if such fin whale genome annotation was available, using an assembly of less quality could negatively impact our results, especially the accuracy of the deleterious variation analyses, and given the results of our reanalysis comparing genotyping and diversity estimates it seems that using the fin whale genome assembly and annotation will not make a significant difference for our main results and conclusions.” Supplemental text lines 217 – 227.

And there seems to be a hiccup at line 678.

R3. We have solved this error in the tracked changes text.

Reference:

Beichman, A. C., Kalhori, P., Kyriazis, C. C., DeVries, A. A., Nigenda-Morales, S., Heckel, G., Schramm, Y., Moreno-Estrada, A., Kennett, D. J., Hylkema, M., Bodkin, J., Koepfli, K., Lohmueller, K. E., & Wayne, R. K. (2022). Genomic analyses reveal range-wide devastation of sea otter populations. *Molecular Ecology*, *mec.16334*. <https://doi.org/10.1111/mec.16334>